# LOW VARIANCE: A BOTTLENECK IN DIFFUSION-BASED GRAPH IMPUTATION

## ABSTRACT

In this paper, we tackle learning tasks on graphs with missing features, improving the applicability of graph neural networks to real-world graph-structured data. Existing imputation methods based upon graph diffusion produce channels that have nearly identical values within each channel, and these low-variance channels contribute very little to performance in graph learning tasks. To prevent diffusion-based imputation from producing low-variance channels, we introduce synthetic features that address the cause of the production, thereby increasing variance in low-variance channels. Since the synthetic features prevent diffusion-based imputation models from generating meaningless feature values shared across all nodes, our synthetic feature propagation design prevents significant performance degradation, even under extreme missing rates. Extensive experiments demonstrate the effectiveness of our scheme across various graph learning tasks with missing features, ranging from low to extremely high missing rates. Moreover, we provide empirical evidence and theoretical proof that validate the low-variance problem.

## 1 INTRODUCTION

Graph neural networks (GNNs) have achieved significant successes in graph learning tasks such as node classification (Kipf & Welling, 2016a; Veličković et al., 2017) and link prediction (Kipf & Welling, 2016b; Salha et al., 2019). Since a wide range of data contains entities with relations, these data can be represented in graphs and many problems are formulated as graph learning tasks (Wu et al., 2022; Liao et al., 2021). However, real-world graph-structured data often include missing features for various reasons (*e.g.*, private information in social networks and measurement failure), which hinders GNNs from being directly applied to real-world data. Therefore, applying GNNs to graphs with missing features has received great attention as a task termed graph learning task with missing features (Chen et al., 2020; Taguchi et al., 2021).

Recently, the diffusion-based imputation approaches (Rossi et al., 2022; Um et al., 2023), which impute missing features by diffusing observed features along edges in a channel-wise manner, have shown promising results. The imputed features with the features diffused from observed features can provide sufficient information for the downstream graph learning tasks (Um et al., 2023). The diffusion-based methods demonstrate the following two advantages against conventional neural-network-based imputation methods (Monti et al., 2017; Chen et al., 2020): 1) superior performance and 2) fast imputation without learnable parameters.

In this paper, we unveil an inherent limitation of the diffusion-based methods: when all observed features within a channel have almost the same values, the diffusion process fills all missing features in the channel with nearly the same values. We refer to such channels having nearly the same values across the nodes (*i.e.*, low-variance) as *low-variance channels*. As illustrated in Figure 1(a), we observe that in outputs of state-of-the-art diffusion-based methods (Rossi et al., 2022; Um et al., 2023), the majority of channels tend to be low-variance channels. We further provide theoretical proof that diffusion-based methods produce a zero-variance channel when observed features within a channel have the same values. Having almost identical values across the nodes, the low-variance channels contribute minimally to graph learning tasks that demand distinct representations of nodes or node pairs as shown in Figure 1(b).

To address the aforementioned low-variance channel issue, we propose a novel diffusion-based imputation scheme called Feature Imputation with Synthetic Features (FISF). Specifically, FISF consists of two diffusion stages. First, to identify low-variance channels, FISF imputes missing features

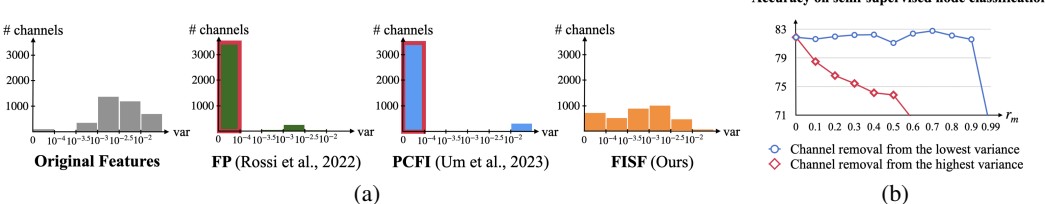

Figure 1: **(a)** Distributions of variance for each feature channel. The distributions for imputation methods are calculated from imputed matrices for the CiteSeer dataset with $99.5\%$ missing features. While existing diffusion-based imputation methods (FP and PCFI) produce outputs with many low-variance channels (outlined in red), our FISF (Feature Imputation with Synthetic Feature) effectively addresses the problem of low-variance channels. **(b)** Accuracy (%) on semi-supervised node classification tasks while progressively excluding channels from the original feature matrix. The accuracy persists despite an increasing removal proportion when channels are excluded in ascending order of variance, starting from the lowest (The blue line). However, removing channels starting from the highest variance leads to significant performance degradation.

using existing diffusion-based methods (Rossi et al., 2022; Um et al., 2023). In each identified low-variance channel, FISF removes all the imputed features and generates a synthetic feature by injecting random noise into a randomly chosen node. This random noise injection to low-variance channels makes these channels deviate from low variances, which increases the distinctiveness of the final imputed features. Lastly, FISF diffuses both observed and synthetic features, making the final imputed features. Despite its simplicity, we verify that FISF provides surprisingly effective imputed features as shown in Figure 1(a), allowing GNN models to achieve remarkable performance gains in downstream graph learning tasks.

Our key contributions are summarized as follows: 1) We discover a phenomenon wherein diffusion-based imputation methods result in low-variance channels in their outputs, supported by both empirical and theoretical evidence. 2) We propose FISF, a novel diffusion-based imputation method that tackles the issue of low-variance channels by leveraging synthetic features. To the best of our knowledge, this work is the first attempt to use synthetic features for imputation. 3) Through extensive experiments, we demonstrate that our FISF effectively removes low-variance channels in output matrices, contributing to significant performance gains on both semi-supervised node classification and link prediction tasks under various missing feature settings.

## 2 RELATED WORK

### 2.1 LEARNING ON GRAPHS WITH MISSING FEATURES

Dealing with missing data has long been an active research field in machine learning (Allison, 2009; Troyanskaya et al., 2001). Methods for handling missing data in graph-structured data can be categorized into three groups.

(i) *GNN Architecture.* Several methods propose new GNN architectures to perform learning tasks on graphs with missing features. GCN for missing features (GCNMF) (Taguchi et al., 2021) combines a GCN (Kipf & Welling, 2016a) layer with a Gaussian mixture model that represents missing features. Jiang & Zhang (2020) develops a message passing layer that aggregates only known features. Graph feature neural network (GRAFENNE) (Gupta et al., 2023) consists of three-phase message-passing layers to address heterogeneous and dynamic features. However, these methods, with their specially designed layers, cannot take advantage of the off-the-shelf GNN models.

(ii) *Reconstruction.* Reconstruction-based methods train models by minimizing the reconstruction error between observed features and their reconstructed values. Recurrent Multi-Graph CNN (RMGCNN) leverages recurrent neural networks to complete a feature matrix (Monti et al., 2017). Structure-attribute-transformer (SAT) (Chen et al., 2020) models the joint distribution of graph structures and node features. Max-entropy graph autoencoder (MEGAE) (Gao et al., 2023) maximizes the entropy of latent features in autoencoders to alleviate the spectral concentration problem. While these methods aim to accurately reconstruct missing features, achieving accurate reconstructed features does not necessarily guarantee high performance in downstream tasks (Um et al., 2023).

(iii) *Diffusion.* In this paper, diffusion-based imputation refers to an approach that imputes missing features by diffusing known features without trainable parameters. Diffusion-based imputation is based on feature homophily, the tendency that features of connected nodes are often similar on a graph. While preserving observed features, missing features are updated by repeatably aggregating features from neighboring nodes. Feature propagation (FP) (Rossi et al., 2022) is pioneering work, which iteratively propagates known features in a channel-wise manner and fills in missing features. Pseudo-confidence-based feature imputation (PCFI) (Um et al., 2023) calculates pseudo-confidence of each feature value and leverages pseudo-confidence as the importance of feature values during diffusion. These diffusion-based techniques have been favored due to their effectiveness at high rates of missing. However, these techniques tend to make missing features very similar to each other when a few observed features are highly similar, resulting in minimal feature differences between nodes. Our approach encourages distinct features between nodes, which can further enhance the performance of downstream GNNs in graph learning tasks.

## 2.2 Distance Encoding

To spread synthetic features widely, we assign different importance to each feature based on distance encoding. Distance encoding is a technique that utilizes graph-distance measures (*e.g.*, shortest path distance, generalized PageRank scores (Li et al., 2019)) measured between a node and a designated node set. You et al. (2019) proposes an aggregation scheme using the computed distance of a given node from sampled anchor node sets. Zhang & Chen (2018) and Li et al. (2020) leverage encoded distance as extra node features for link prediction. Position-aware graph neural network (P-GNN) (Zhang et al., 2021) unifies several techniques including distance encoding into a labeling trick.

## 3 Notation and Problem Definition

**Notation.** An undirected connected graph can be represented as $\mathcal{G} = (\mathcal{V}, \mathcal{E}, \mathbf{A})$ where $\mathcal{V} = \{v_1, \ldots, v_N\}$ is the set of $N$ nodes, $\mathcal{E}$ is the edge set, and $\mathbf{A} \in \{0,1\}^{N \times N}$ is an adjacency matrix. $\mathbf{X} = [x_{i,a}] \in \mathbb{R}^{N \times F}$ denotes a node feature matrix where $F$ is the number of feature channels and $x_{i,a}$ represents the $a$-th channel feature value of $v_i$.

Let $d(v_i, v_j | \mathbf{A})$ be the shortest path distance between the $i$-th node and the $j$-th node on $\mathcal{G}$ with $\mathbf{A}$. Then, we define a function $d_{set}(\cdot)$ as $d_{set}(v_i | \mathcal{V}', \mathbf{A}) = \min_{v_j \in \mathcal{V}'} d(v_i, v_j | \mathbf{A})$ where $\mathcal{V}' \subseteq \mathcal{V}$. That is, we use $d_{set}(v_i | \mathcal{V}', \mathbf{A})$ to denote the shortest path distance between the $i$-th node and its nearest node in a node set $\mathcal{V}' \subseteq \mathcal{V}$ on $\mathcal{G}$ with $\mathbf{A}$.

Partially known (observed) features mean that $\mathbf{X}$ has missing elements. $\mathcal{V}_k^{(a)}$ denotes a set of nodes whose $a$-th channel feature values are known. $\mathcal{V}_u^{(a)}$ denotes a set of nodes whose $a$-th channel feature values are unknown (missing) (*i.e.*, $\mathcal{V}_u^{(a)} = \mathcal{V} \setminus \mathcal{V}_k^{(a)}$). We refer to $\mathcal{V}_k^{(a)}$ and $\mathcal{V}_u^{(a)}$ as source nodes and missing nodes, respectively. By rearranging the whole nodes based on whether the feature value is known or not for each channel, the whole features and the adjacency matrix for the $a$-th channel can be written as

$$\mathbf{x}^{(a)} = \begin{bmatrix} \mathbf{x}_k^{(a)} \\ \mathbf{x}_u^{(a)} \end{bmatrix}, \qquad \mathbf{A}^{(a)} = \begin{bmatrix} \mathbf{A}_{kk}^{(a)} & \mathbf{A}_{ku}^{(a)} \\ \mathbf{A}_{uk}^{(a)} & \mathbf{A}_{uu}^{(a)} \end{bmatrix}, \tag{1}$$

where $\mathbf{x}^{(a)}$, $\mathbf{x}_k^{(a)}$, and $\mathbf{x}_u^{(a)}$ are column vectors for the $a$-th channel. $\mathbf{A}^{(a)}$ and $\mathbf{A}$ represent the same graph structure although the node order of $\mathbf{A}^{(a)}$ is rearranged from $\mathbf{A}$. We use $\mathbf{B}_{:,z}$ to denote the $z$-th column of a matrix $\mathbf{B}$.

**Problem definition.** We tackle a problem of graph learning tasks containing missing features, where our goal is to minimize performance degradation in downstream learning tasks despite high rates of missing features. Formally, graph learning tasks containing missing features can be expressed as

$$\hat{\mathbf{Y}} = f(\{\mathbf{x}_k^{(a)}\}_{a=1}^F, \mathbf{A}) \tag{2}$$

where $\hat{\mathbf{Y}}$ denotes a prediction for desired output of a given task. Here, $f$ is a function to find in the problem. We decompose $f$ into two steps as $f = g_\theta \circ h$. Here, $h$ is a feature imputation scheme and $g_\theta$ is an off-the-shelf GNN model using a full feature matrix obtained via $h$.

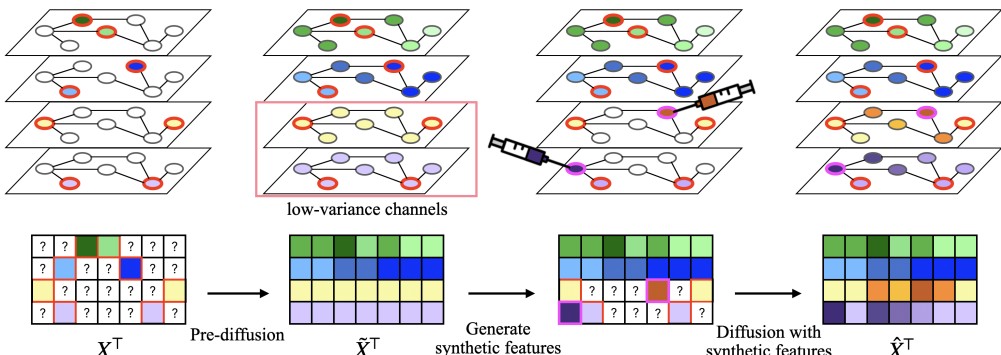

Figure 2: A brief overview of feature imputation with synthetic features (FISF). First, pre-diffusion constructs a full feature matrix $\tilde{\mathbf{X}}$ by imputing missing features via channel-wise diffusion. Then, for each low-variance channel in $\tilde{\mathbf{X}}$, we inject one synthetic feature into a randomly chosen node from nodes with missing features. Finally, diffusion with synthetic features produces $\hat{\mathbf{X}}$ which is a final output of FISF. $\hat{\mathbf{X}}$ is fed to a downstream GNN which performs a given graph learning task.

## 4 PROPOSED METHOD

### 4.1 OVERVIEW OF FISF

We present an imputation scheme called feature imputation with synthetic features (FISF), which minimizes performance degradation in graph learning tasks despite high rates of missing features. Figure 2 shows a brief overview of FISF which consists of two diffusion stages: *pre-diffusion* and *diffusion with synthetic features*. Using a pre-imputed feature matrix obtained via pre-diffusion (see Section 4.2), we calculate the variance of features for each channel. We then create a synthetic feature in each low-variance channel (see Section 4.3). The second diffusion stage updates the features in low-variance channels by spreading the synthetic features widely (see Section 4.4). The stage produces a final output feature matrix of FISF, which is fed to $g_\theta$ to perform downstream tasks.

### 4.2 PRE-DIFFUSION

We adopt channel-wise inter-node diffusion in PCFI (Um et al., 2023) as pre-diffusion. It is noteworthy that FP (Rossi et al., 2022) can be also used for pre-diffusion (See Appendix C.4). For notational convenience, we temporarily rearrange all nodes in a channel-wise manner as described in Section 3. Specifically, for the $a$-th channel, we reorder the nodes in the order of $\mathcal{V}_k^{(a)}$ and $\mathcal{V}_u^{(a)}$, *i.e.*, $\mathbf{x}^{(a)}$ and $\mathbf{A}^{(a)}$ are made by reordering $\mathbf{A}$. After the diffusion is completed, we restore the node ordering to the original one.

The channel-wise inter-node diffusion calculates and utilizes pseudo-confidence (PC) (Um et al., 2023), which acts as the importance of each feature value during the diffusion. We use $\mathbf{S}_{i,a}$ to denote the shortest path distance between the $i$-th node and its nearest source node for the $a$-th channel, *i.e.*, $\mathbf{S}_{i,a} = d_{set}(v_i | \mathcal{V}_k^{(a)}, \mathbf{A}^{(a)})$. We let $\tilde{X}$ be a pre-imputed feature matrix via pre-diffusion. Then, following Um et al. (2023), PC ($\xi_{i,a}$) of $\tilde{x}_{i,a}$ is assigned by $\xi_{i,a} = \alpha^{\mathbf{S}_{i,a}} (0 < \alpha < 1)$ where $\alpha$ is a hyperparameter. Thereafter, the transition matrix for the pre-diffusion is built by a weighted adjacency matrix $\mathbf{W}^{(a)} \in \mathbb{R}^{N \times N}$ given by

$$\mathbf{W}_{i,j}^{(a)} = \begin{cases} \xi_{j,a}/\xi_{i,a} & \text{if } \mathbf{A}_{i,j}^{(a)} = 1 \\ 0 & \text{if } \mathbf{A}_{i,j}^{(a)} = 0, \end{cases} \tag{3}$$

where $\mathbf{W}_{i,j}^{(a)}$ takes a role of message passing strength from the $j$-th node to the $i$-th node in the pre-diffusion. For a row-stochastic transition matrix, we normalize $\mathbf{W}^{(a)}$ to $\overline{\mathbf{W}}^{(a)} = (\mathbf{D}^{(a)})^{-1} \mathbf{W}^{(a)}$ where $\mathbf{D}^{(a)}$ is a diagonal matrix with diagonal entries $\mathbf{D}_{i,i}^{(a)} = \sum_j \mathbf{W}_{i,j}$. Then, to preserve the known features $\mathbf{x}_k^{(a)}$ during the pre-diffusion, we replace the first $|\mathcal{V}_k^{(a)}|$ rows in $\overline{\mathbf{W}}$ with one-hot vectors indicating $\mathcal{V}_k^{(a)}$. As a result of the replacement, we attain the pre-diffusion transition matrix

$\widetilde{\mathbf{W}}^{(a)}$ expressed by

$$\widetilde{\mathbf{W}}^{(a)} = \begin{bmatrix} \mathbf{I}_{kk} & \mathbf{0}_{ku} \\ \overline{\mathbf{W}}_{uk}^{(a)} & \overline{\mathbf{W}}_{uu}^{(a)} \end{bmatrix}, \tag{4}$$

where $\mathbf{I}_{kk} \in \mathbb{R}^{|\mathcal{V}_k^{(a)}| \times |\mathcal{V}_k^{(a)}|}$ is an identity matrix and $\mathbf{0}_{ku} \in \mathbb{R}^{|\mathcal{V}_k^{(a)}| \times |\mathcal{V}_u^{(a)}|}$ is a zero matrix.

The pre-diffusion is implemented by iterative propagation steps using $\widetilde{\mathbf{W}}^{(a)}$ as

$$\tilde{\mathbf{x}}^{(a)}(t) = \widetilde{\mathbf{W}}^{(a)} \tilde{\mathbf{x}}^{(a)}(t-1), \ \ t = 1, \cdots, K;$$

$$\tilde{\mathbf{x}}^{(a)}(0) = \begin{bmatrix} \mathbf{x}_k^{(a)} \\ \mathbf{0}_u \end{bmatrix}, \tag{5}$$

where $\tilde{\mathbf{x}}^{(a)}(t)$ is an imputed feature vector after $t$ propagation steps and $\mathbf{0}_u$ is a zero vector with a length of $|\mathcal{V}_u^{(a)}|$. After $K$ propagation steps, we obtain $\tilde{\mathbf{x}}^{(a)}(K)$. As $K \to \infty$, the recursion converges and $\tilde{\mathbf{x}}^{(a)}(K)$ reaches a steady state (see the proof in Appendix A)). Based on the proof that initial values for $\mathbf{x}_u^{(a)}$ do not affect the steady state, we initialize $\mathbf{x}_u^{(a)}$ with zeros (*i.e.*, $\mathbf{0}_u$). We use $\tilde{\mathbf{x}}^{(a)}(K)$ with large enough $K$ to approximate the steady state.

We rearrange $\{\tilde{\mathbf{x}}^{(a)}(K)\}_{a=1}^F$ in the original order to reorder the nodes considering synthetic features in the second diffusion stage. Then, by stacking the originally ordered vectors in $\{\tilde{\mathbf{x}}^{(a)}(K)\}_{a=1}^F$ along the channels, we obtain a pre-imputed feature matrix $\tilde{\mathbf{X}}$ which is an output of the pre-diffusion.

## 4.3 SYNTHETIC FEATURE GENERATION

When all given known features in the $a$-th channel (*i.e.*, elements in $\mathbf{x}_k^{(a)}$) have the same value $c$, $\lim_{t \to \infty} \tilde{\mathbf{x}}^{(a)}(t)$ becomes a vector where all elements are $c$ (see the proof in Appendix B)). We refer to a channel with the same or nearly the same feature values as a *low-variance channel*. The low-variance channel does not contribute to distinguishing nodes. In semi-supervised node classification, distinctive node representations are crucial to classify nodes into multiple classes. In the case of link prediction, the same representation across nodes also makes the representations of node pairs the same. Therefore, we aim to make imputed features in that channel become distinctive across nodes by injecting a synthetic feature that acts as a known feature.

We first identify low-variance channels to inject synthetic features. We calculate the variance of $\tilde{\mathbf{X}}_{:,a}$ (*i.e.*, pre-imputed feature values in the $a$-th channel) for all $a \in \{1, \ldots, F\}$. Then $r\%$ of channels are selected in order of lowest to highest variance, where $r$ is a hyperparameter between 0 and 100. $\mathbb{F}_l$ denotes the set of low-variance channel indices. For each channel in $\mathbb{F}_l$, we randomly choose one node with a missing feature to inject a synthetic feature. For a selected node $v_s^{(b)}$ in a channel $b \in \mathbb{F}_l$, we inject a synthetic feature with randomly sampled value $\mathbf{x}_s^{(b)}$ from a uniform distribution on $[0, 1]$. Consequently, $|\mathbb{F}_l|$ number of synthetic feature values are injected and $\{(v_s^{(b)}, \mathbf{x}_s^{(b)})\}_{b \in \mathbb{F}_l}$ is combined with the result of pre-diffusion ($\tilde{\mathbf{X}}$) for the second diffusion stage called diffusion with synthetic features.

## 4.4 DIFFUSION WITH SYNTHETIC FEATURES

Diffusion with synthetic features (DSF) produces $\hat{\mathbf{X}} = [\hat{x}_{i,a}] \in \mathbb{R}^{N \times F}$ which is a final output of FISF. DSF receives $\tilde{\mathbf{X}}$ from the pre-diffusion and $\{(v_s^{(b)}, \mathbf{x}_s^{(b)})\}_{b \in \mathbb{F}_l}$. Then DSF updates $\tilde{\mathbf{X}}$ by replacing features in the low-variance channels (*i.e.*, $\tilde{\mathbf{X}}_{:,b}$ for all $b \in \mathbb{F}_l$). The purpose of DSF is to increase the variance of low-variance channels by using synthetic features.

DSF treats a synthetic feature $\mathbf{x}_s^{(b)}$ as known features $\mathbf{x}_k^{(b)}$ during diffusion. Then the updated known node set becomes $\mathcal{V}_{k^*}^{(b)} = \mathcal{V}_k^{(b)} \cup \{v_s^{(b)}\}$. Thus the updated unknown node set becomes $\mathcal{V}_{u^*}^{(b)} = \mathcal{V}_u^{(b)} \setminus \{v_s^{(b)}\}$. That is, $v_s^{(b)}$ is moved from $\mathcal{V}_u^{(b)}$ to $\mathcal{V}_{k^*}^{(b)}$. Similar to pre-diffusion, we first temporarily reorder all the nodes in the order of $\mathcal{V}_{k^*}^{(b)}$ and $\mathcal{V}_{u^*}^{(b)}$. By reordering, features and the adjacency matrix in the $b$-th channel in $\mathbb{F}_l$ can be expressed as

$$\mathbf{x}^{(b)} = \begin{bmatrix} \mathbf{x}_{k^*}^{(b)} \\ \mathbf{x}_{u^*}^{(b)} \end{bmatrix}, \qquad \mathbf{A}^{(b)} = \begin{bmatrix} \mathbf{A}_{k^*k^*}^{(b)} & \mathbf{A}_{k^*u^*}^{(b)} \\ \mathbf{A}_{u^*k^*}^{(b)} & \mathbf{A}_{u^*u^*}^{(b)} \end{bmatrix}, \tag{6}$$

where $\mathbf{x}_{k^*}^{(b)}$ and $\mathbf{x}_{u^*}^{(b)}$ are column vectors and $\mathbf{x}_{k^*}^{(b)}$ contains $\mathbf{x}_s^{(b)}$. The length of $\mathbf{x}_{k^*}^{(b)}$ and $\mathbf{x}_{u^*}^{(b)}$ are $|\mathcal{V}_k^{(b)}| + 1$ and $|\mathcal{V}_u^{(b)}| - 1$, respectively.

The preparations above are the same as the pre-diffusion, except for assuming $\mathbf{x}_s^{(b)}$ as a known feature. However, simply diffusing features of $\mathcal{V}_{k^*}^{(b)}$ as pre-diffusion results in $\mathbf{x}_s^{(b)}$ influencing only its surroundings. This is because not only $\mathbf{x}_s^{(b)}$ but also known features with nearly the same values diffuse. For example, if a given graph has $10,000$ nodes and $90\%$ features are missing in the $b$-th channel, there exist $1,000$ known features with nearly the same feature values in the channel. Known features spread to their surrounding features through diffusion and make the surrounding features be similar to their own value. Thus, it is hard for $\mathbf{x}_s^{(b)}$ to exert a wide influence across nodes. This issue hinders the channel from deviating from a low variance since most of the features become nearly the same value.

To overcome the issue, we design DSF to give more influence to synthetic features than that of known features. For the wide diffusion of $\mathbf{x}_s^{(b)}$, we leverage the shortest path distance from $v_s^{(b)}$. We measure the shortest path distance from $v_s^{(b)}$ to all nodes in $\mathcal{V}$. Formally, we use $\mathbf{S}_{i,b}^s$ to denote $d(v_i, v_s^{(b)} | \mathbf{A}^{(b)})$ and measure $\mathbf{S}_{i,b}^s$ for all $v_i \in \mathcal{V}$.

Then the PC $\xi_{i,a}^s$ of $\hat{x}_{i,a}$ is computed based on the shortest path distance from only the synthetic node $v_s^{(b)}$, not from the whole known nodes. That is, $\xi_{i,a}^s$ is defined by $\xi_{i,a}^s = \beta^{\mathbf{S}_{i,a}^s} (0 < \beta < 1)$ where $\beta$ is a hyperparameter. As $v_i$ is positioned closer to $v_s^{(b)}$, $\xi_{i,a}^s$ increases. We also use usual PC ($\xi_{i,b}^*$) based on distances from the whole known nodes $\mathcal{V}_{k^*}^{(b)}$ containing $v_s^{(b)}$. We calculate $\mathbf{S}_{i,b}^* = d_{set}(v_i | \mathcal{V}_{k^*}^{(b)}, \mathbf{A}^{(b)})$ and obtain PC calculated by $\xi_{i,b}^* = \alpha^{\mathbf{S}_{i,b}^*} (0 < \alpha < 1)$. While both $\xi_{i,b}$ and $\xi_{i,b}^*$ play a role as the importance of each feature value, $\xi_{i,b}$ is determined by the distance from only synthetic node $v_s^{(b)}$ in contrast to $\xi_{i,b}^*$ considering the distances from whole known nodes $\mathcal{V}_{k^*}^{(b)}$. Using the PCs, we define a weighted adjacency matrix $\mathbf{M}^{(b)} \in \mathbb{R}^{N \times N}$ by

$$\mathbf{M}_{i,j}^{(b)} = \begin{cases} \dfrac{\xi_{j,b}^*}{\xi_{i,b}^*} \cdot \dfrac{\xi_{j,b}^s}{\xi_{i,b}^s} & \text{if } \mathbf{A}_{i,j}^{(b)} = 1 \\ 0 & \text{if } \mathbf{A}_{i,j}^{(b)} = 0. \end{cases} \tag{7}$$

$\mathbf{M}_{i,j}^{(b)}$ is the strength of a message passing from the $j$-th node to the $i$-th node in the DSF.

The term $\xi_{j,b}^* / \xi_{i,b}^*$ strengthens a message passing from a high-PC feature to a low-PC feature as in the pre-diffusion (see Eq. 3). However, different from the pre-diffusion, the synthetic feature of $v_s^{(b)}$ is considered as one of the nodes in $\mathcal{V}_k^{(b)}$. Thus the influence of the synthetic feature is very weak compared to that of the many observed similar features. To widely spread the synthetic feature, we introduce the term $\xi_{j,b}^s / \xi_{i,b}^s$, which strengthens a message passing from a feature of a node near $v_s^{(b)}$ to a feature of a node far from $v_s^{(b)}$. This term makes the synthetic feature spread widely compared to observed features. The design goals of the two terms naturally combine through multiplication in Eq. 7. $\xi_{i,b}^*$ is 1 for both $v \in \mathcal{V}_k^{(b)}$ and $\mathbf{x}_s^{(b)}$. However, $\xi_{i,b}^s$ is 1 for $\mathbf{x}_s^{(b)}$ while it is at most $\beta$ for $v \in \mathcal{V}_k^{(b)}$. Therefore, in the second stage diffusion, the synthetic feature has a greater influence than observed features.

To construct a transition matrix, we prepare a row-stochastic matrix by normalizing $\mathbf{M}^{(b)}$ to $\overline{\mathbf{M}}^{(b)} = (\mathbf{D}'^{(b)})^{-1} \mathbf{W}^{(b)}$ where $\mathbf{D}'^{(b)}$ is a diagonal matrix with $\mathbf{D}_{ii}'^{(b)} = \sum_j \mathbf{M}_{i,j}$. Then, we replace the first $|\mathcal{V}_{k^*}^{(b)}|$ rows in $\overline{\mathbf{M}}$ with one-hot vectors representing $\mathcal{V}_{k^*}^{(b)}$ to preserve $\mathbf{x}_{k^*}^{(b)}$ including $\mathbf{x}_s^{(b)}$. By the replacement, we obtain a DSF transition matrix $\widetilde{\mathbf{M}}^{(b)}$ as follows:

$$\widetilde{\mathbf{M}}^{(b)} = \begin{bmatrix} \mathbf{I}_{k^* k^*} & \mathbf{0}_{k^* u^*} \\ \mathbf{M}_{u^* k^*}^{(b)} & \overline{\mathbf{M}}_{u^* u^*}^{(b)} \end{bmatrix}, \tag{8}$$

where $\mathbf{I}_{k^* k^*} \in \mathbb{R}^{|\mathcal{V}_{k^*}^{(b)}| \times |\mathcal{V}_{k^*}^{(b)}|}$ is an identity matrix and $\mathbf{0}_{k^* u^*} \in \mathbb{R}^{|\mathcal{V}_{k^*}^{(b)}| \times |\mathcal{V}_{u^*}^{(b)}|}$ is a zero matrix.

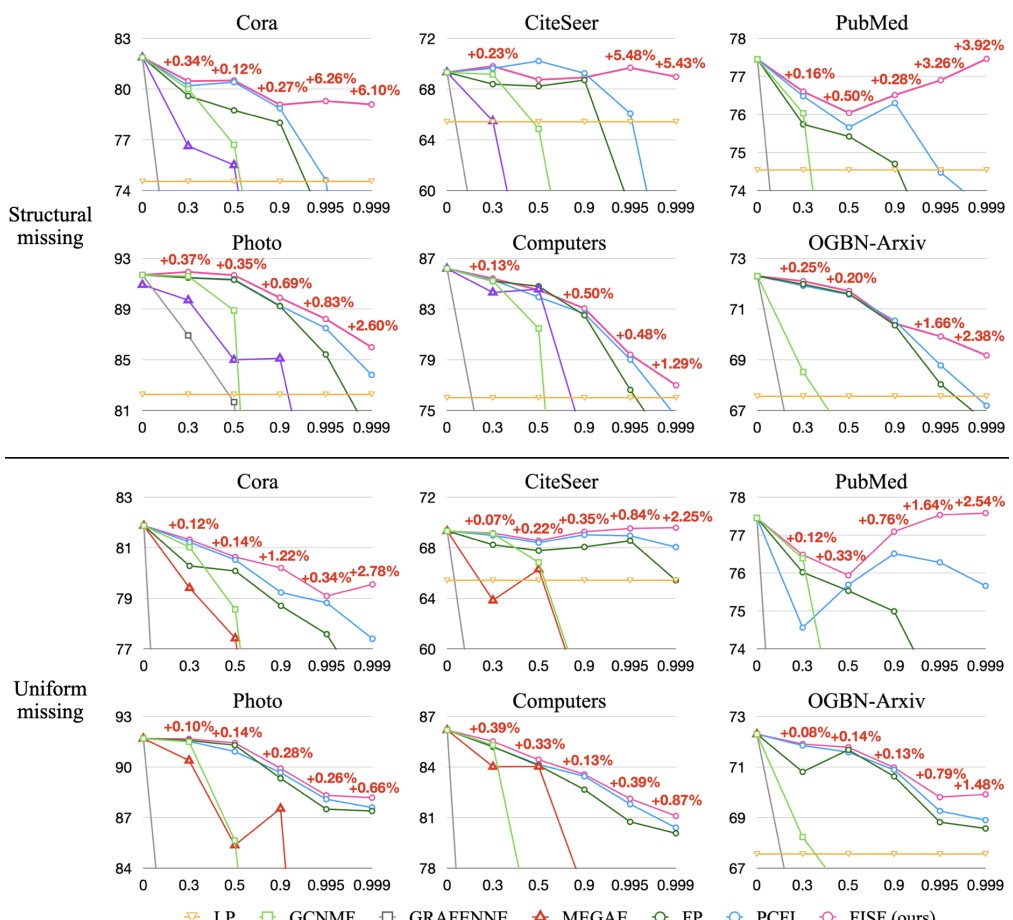

Figure 3: Accuracy (%) on semi-supervised node classification tasks under structural-missing and uniform-missing settings with $r_m \in \{0.3, 0.5, 0.9, 0.995, 0.999\}$. Figures highlighted in red indicate performance improvements over the most competitive baseline across each setting. Cases where accuracy cannot be measured due to out-of-memory errors are not included.

We define diffusion with synthetic features (DSF) by

$$
\begin{aligned}
\hat{\mathbf{x}}^{(b)}(t) &= \widetilde{\widetilde{\mathbf{M}}}^{(b)}\hat{\mathbf{x}}^{(b)}(t-1), \ \ t = 1, \cdots, K; \\
\hat{\mathbf{x}}^{(b)}(0) &= \begin{bmatrix} \mathbf{x}_{k^*}^{(b)} \\ \mathbf{0}_{u^*} \end{bmatrix},
\end{aligned}
\tag{9}
$$

where $\hat{\mathbf{x}}^{(b)}(t)$ denotes an imputed feature vector after $t$ propagation steps and $\mathbf{0}_{u^*}$ denotes a zero vector of the same length as $|\mathcal{V}_{u^*}^{(b)}|$. As $K \to \infty$, $\hat{\mathbf{x}}^{(b)}(K)$ converges (see the proof in Appendix A). With sufficiently large $K$, we approximate the steady state $\lim_{t \to \infty} \hat{\mathbf{x}}^{(b)}(t)$ to $\hat{\mathbf{x}}^{(b)}(K)$. We perform DSF in the $b$-th channel for all $b \in \mathbb{F}_l$ and obtain $\{\hat{\mathbf{x}}^{(b)}(K)\}_{b \in \mathbb{F}_l}$. Since vectors in $\{\hat{\mathbf{x}}^{(b)}(K)\}_{b \in \mathbb{F}_l}$ have different ordering from the original one, we restore ordering of all the vectors according to the original order. To construct $\hat{\mathbf{X}} \in \mathbb{R}^{N \times F}$, we prepare $\tilde{\mathbf{X}} \in \mathbb{R}^{N \times F}$ from the pre-diffusion and replace $\tilde{\mathbf{X}}_{:,b}$ for all $b \in \mathbb{F}_l$ with the corresponding vector in $\{\hat{\mathbf{x}}^{(b)}(K)\}_{b \in \mathbb{F}_l}$. The feature matrix with the replaced columns is $\hat{\mathbf{X}}$, a final output of FISF. $\hat{\mathbf{X}}$ is fed to a GNN to perform a given task.

## 5 Experiments

We perform comparative evaluation of FISF against state-of-the-art methods on two main graph learning tasks: semi-supervised node classification and link prediction.

Table 1: Performance on semi-supervised node classification tasks at $r_m = 0.995$, measured by accuracy (%). Standard deviation errors are given. OOM denotes an out-of-memory error.

### Structural missing

| Method | CORA | CITESEER | PUBMED | PHOTO | COMPUTERS | OGBN-ARXIV |
|--------|------|----------|--------|-------|-----------|------------|
| Full features | $81.87 \pm 1.59$ | $69.32 \pm 0.57$ | $77.45 \pm 2.17$ | $91.69 \pm 0.78$ | $86.19 \pm 0.78$ | $72.30 \pm 0.10$ |
| LP | $74.54 \pm 1.79$ | $65.42 \pm 1.80$ | $71.67 \pm 4.94$ | $82.27 \pm 2.72$ | $76.01 \pm 1.84$ | $67.56 \pm 0.00$ |
| GCNMF | $31.33 \pm 2.73$ | $24.84 \pm 2.44$ | $40.48 \pm 0.53$ | $25.60 \pm 0.17$ | $37.21 \pm 0.08$ | $9.00 \pm 6.27$ |
| GRAFENNE | $20.2 \pm 10.98$ | $17.58 \pm 2.94$ | $33.12 \pm 2.43$ | $21.10 \pm 17.39$ | $16.31 \pm 11.84$ | $13.66 \pm 12.23$ |
| MEGAE | $38.78 \pm 4.76$ | $32.94 \pm 4.08$ | OOM | $68.90 \pm 9.46$ | $42.37 \pm 5.03$ | OOM |
| FP | $71.86 \pm 2.82$ | $58.61 \pm 1.74$ | $71.96 \pm 3.06$ | $85.42 \pm 3.16$ | $76.62 \pm 1.94$ | $68.03 \pm 0.52$ |
| PCFI | $74.62 \pm 1.78$ | $66.06 \pm 3.26$ | $74.47 \pm 2.54$ | $87.49 \pm 1.50$ | $79.02 \pm 1.22$ | $68.78 \pm 0.25$ |
| FISF | $\mathbf{79.29 \pm 1.72}$ | $\mathbf{69.68 \pm 2.47}$ | $\mathbf{76.90 \pm 1.50}$ | $\mathbf{88.22 \pm 0.79}$ | $\mathbf{79.40 \pm 1.11}$ | $\mathbf{69.92 \pm 0.17}$ |

### Uniform missing

| Method | CORA | CITESEER | PUBMED | PHOTO | COMPUTERS | OGBN-ARXIV |
|--------|------|----------|--------|-------|-----------|------------|
| Full features | $81.87 \pm 1.59$ | $69.32 \pm 0.57$ | $77.45 \pm 2.17$ | $91.69 \pm 0.78$ | $86.19 \pm 0.78$ | $72.30 \pm 0.10$ |
| LP | $74.54 \pm 1.79$ | $65.42 \pm 1.80$ | $71.67 \pm 4.94$ | $82.27 \pm 2.72$ | $76.01 \pm 1.84$ | $67.56 \pm 0.00$ |
| GCNMF | $34.01 \pm 8.08$ | $29.71 \pm 5.12$ | $40.08 \pm 0.45$ | $25.59 \pm 0.16$ | $37.20 \pm 0.08$ | $5.86 \pm 0.00$ |
| GRAFENNE | $20.55 \pm 13.65$ | $19.32 \pm 7.42$ | $34.75 \pm 4.26$ | $29.96 \pm 20.92$ | $21.74 \pm 15.94$ | $15.52 \pm 11.70$ |
| MEGAE | $46.13 \pm 9.06$ | $34.32 \pm 7.65$ | OOM | $55.31 \pm 10.37$ | $41.02 \pm 4.05$ | OOM |
| FP | $77.58 \pm 1.98$ | $68.55 \pm 2.33$ | $72.62 \pm 4.18$ | $87.50 \pm 1.49$ | $80.75 \pm 0.70$ | $68.82 \pm 0.07$ |
| PCFI | $78.82 \pm 1.48$ | $68.94 \pm 1.95$ | $76.28 \pm 2.52$ | $88.09 \pm 1.41$ | $81.80 \pm 0.71$ | $69.26 \pm 0.17$ |
| FISF | $\mathbf{79.09 \pm 1.73}$ | $\mathbf{69.52 \pm 1.81}$ | $\mathbf{77.53 \pm 1.28}$ | $\mathbf{88.32 \pm 1.37}$ | $\mathbf{82.12 \pm 0.51}$ | $\mathbf{69.81 \pm 0.16}$ |

## 5.1 DATASETS AND BASELINES

**Datasets.** We conduct experiments on graph datasets from two different domains: citation networks (Cora (McCallum et al., 2000), CiteSeer (Giles et al., 1998), PubMed (Sen et al., 2008), and OGBN-Arxiv (Hu et al., 2020)) and recommendation networks (Photo and Computers) (Shchur et al., 2018) from Amazon. Detailed information on the datasets is provided in Appendix E.1.

**Baselines.** We compare FISF with LP (Zhuŕ & GhahramanifH, 2002) and five state-of-the-art methods for graph learning tasks with missing features. (1) LP that does not use any feature propagates partially given labels for semi-supervised node classification. (2) GCNMF (Taguchi et al., 2021) and (3) GRAFENNE (Gupta et al., 2023) are GNN architecture-based methods. (4) MEGAE (Gao et al., 2023) is a reconstruction-based method. (5) FP (Rossi et al., 2022) and (6) PCFI (Um et al., 2023) is diffusion-based methods. Since imputation methods (including MEGAE, FP, PCFI, and FISF) combine with GNNs to perform downstream tasks, we commonly utilize vanilla GCN (Kipf & Welling, 2016a) models for semi-supervised node classification. In link prediction, we commonly utilize graph auto-encoder (GAE) models for the imputation methods.

## 5.2 EXPERIMENTAL SETUP

We follow the missing setting in Um et al. (2023). To evaluate models on graphs containing missing features, we remove a fixed rate (e.g., 90%) of features in the datasets. A missing rate denoted as $r_m$ represents the rate of feature removal. We fill the positions where features are removed with NaN values. We remove features in the following two ways: *structural missing* and *uniform missing*. First, in the case of structural missing, we randomly select nodes at a ratio of $r_m$ from entire nodes and remove all the features of the selected nodes. Second, uniform missing removes randomly selected feature values with a ratio of $r_m$ from a feature matrix $\mathbf{X}$. We report average performance (e.g., accuracy, ROC AUC, and AP) after five runs of experiments under a fixed setting. Therefore, for each missing way, we randomly generate five different binary masks with the same size of $\mathbf{X}$ for each dataset. These masks indicate the locations in $\mathbf{X}$ where features are missing.

For semi-supervised node classification tasks, we randomly create five different training/validation/test node splits for all the datasets except for OGBN-Arxiv which has a fixed split according to the specific criteria. For link prediction tasks, we also randomly create five different training/validation/test edge splits of each dataset. OGBN-Arxiv is excluded from the link prediction tasks due to out-of-memory errors. Grid search is employed to tune $\alpha$, $\beta$, and $\gamma$, the three hyperparameters of FISF. $\alpha$ and $\beta$ are searched within $\{0.1, 0.3, 0.5, 0.7, 0.9\}$. $\gamma$ is chosen from $\{10, 30, 50, 70, 90\}$. For all the methods including FISF, we tune hyperparameters based on validation sets. We utilize the publicly released code for all the baselines. Further implementation details including dataset splits, training details, and baseline implementations are provided in Appendix E.2.

Table 2: Performance on link prediction tasks at $r_m = 0.995$, measured by ROC AUC score (%). Standard deviation errors are given. The best result is highlighted in bold and underlined, while the second-best result is highlighted only in bold. OOM denotes an out-of-memory error.

### Structural missing

| Method | CORA | CITESEER | PUBMED | PHOTO | COMPUTERS |
|---|---|---|---|---|---|
| Full features | $92.20 \pm 0.96$ | $90.55 \pm 1.36$ | $96.41 \pm 0.25$ | $95.70 \pm 0.32$ | $93.71 \pm 0.65$ |
| GCNMF | $67.44 \pm 0.45$ | $68.34 \pm 1.79$ | $\mathbf{87.20 \pm 0.28}$ | $81.00 \pm 0.25$ | $82.92 \pm 0.19$ |
| GRAFENNE | $53.79 \pm 5.26$ | $62.96 \pm 13.82$ | $60.11 \pm 6.10$ | $66.44 \pm 1.74$ | $67.23 \pm 1.71$ |
| MEGAE | $67.13 \pm 0.75$ | $69.34 \pm 2.46$ | OOM | $86.53 \pm 1.97$ | $84.89 \pm 1.77$ |
| FP | $83.85 \pm 1.32$ | $79.83 \pm 2.18$ | $78.54 \pm 1.13$ | $94.25 \pm 0.58$ | $91.27 \pm 0.71$ |
| PCFI | $86.75 \pm 0.84$ | $79.38 \pm 1.81$ | $82.49 \pm 0.82$ | $\mathbf{96.65 \pm 0.25}$ | $94.54 \pm 0.37$ |
| FISF | $\mathbf{87.26 \pm 1.44}$ | $\mathbf{84.12 \pm 1.17}$ | $83.19 \pm 0.78$ | $95.86 \pm 0.21$ | $\mathbf{94.70 \pm 0.30}$ |
| FISF+NIP | $\mathbf{87.16 \pm 1.46}$ | $\mathbf{\underline{84.20 \pm 1.70}}$ | $83.28 \pm 0.42$ | $96.35 \pm 0.18$ | $\mathbf{95.29 \pm 0.32}$ |

### Uniform missing

| Method | CORA | CITESEER | PUBMED | PHOTO | COMPUTERS |
|---|---|---|---|---|---|
| Full features | $92.20 \pm 0.96$ | $90.55 \pm 1.36$ | $96.41 \pm 0.25$ | $95.70 \pm 0.32$ | $93.71 \pm 0.65$ |
| GCNMF | $63.46 \pm 1.04$ | $63.50 \pm 3.40$ | $81.73 \pm 3.13$ | $80.98 \pm 0.17$ | $82.95 \pm 0.11$ |
| GRAFENNE | $68.49 \pm 17.00$ | $61.38 \pm 13.53$ | $65.74 \pm 11.32$ | $68.53 \pm 6.57$ | $70.16 \pm 4.12$ |
| MEGAE | $65.86 \pm 1.22$ | $62.21 \pm 3.18$ | OOM | $84.25 \pm 1.35$ | $84.95 \pm 2.20$ |
| FP | $86.79 \pm 1.36$ | $81.55 \pm 2.30$ | $76.87 \pm 2.89$ | $95.96 \pm 0.17$ | $94.10 \pm 0.33$ |
| PCFI | $87.35 \pm 1.28$ | $82.33 \pm 1.88$ | $84.68 \pm 0.75$ | $\mathbf{97.05 \pm 0.16}$ | $\mathbf{95.62 \pm 0.24}$ |
| FISF | $\mathbf{87.44 \pm 0.80}$ | $\mathbf{\underline{83.45 \pm 2.53}}$ | $\mathbf{85.33 \pm 0.47}$ | $96.64 \pm 0.18$ | $95.13 \pm 0.35$ |
| FISF+NIP | $\mathbf{\underline{87.70 \pm 0.77}}$ | $82.53 \pm 1.94$ | $85.32 \pm 0.48$ | $96.67 \pm 0.21$ | $\mathbf{96.09 \pm 0.24}$ |

## 5.3 SEMI-SUPERVISED NODE CLASSIFICATION RESULTS

To investigate how $r_m$ affects semi-supervised node classification accuracy, we conduct experiments by increasing $r_m$ while keeping all other settings fixed. Figure 3 demonstrates accuracy under structural-missing and uniform-missing settings with varying $r_m$. The accuracy of LP remains consistent since LP does not utilize features. For all methods except for LP, the accuracy tends to decrease as $r_m$ increases. While diffusion-based imputation methods outperform the other methods, FP and PCFI suffer performance degradation as $r_m$ increases. However, FISF shows robust performance despite high $r_m$ regardless of the datasets. Note that FISF using only $0.1\%$ of features (*i.e.*, $r_m = 0.999$) performs similarly to or even outperforms FISF with full features on Cora, Cite-Seer, and PubMed. Furthermore, FISF consistently demonstrates superiority across various missing rates ($r_m$), including low $r_m$, regardless of the missing patterns. The performance gain obtained with FISF diminishes as the missing rate decreases. This is natural since a smaller $r_m$ means fewer missing features to impute, making it difficult to achieve a significant improvement solely through the superiority of the imputation method. Nevertheless, FISF consistently shows its effectiveness at even low $r_m$.

We then conduct experiments to investigate how semi-supervised node classification accuracy varies depending on the missing ways (structural and uniform missing) at the same $r_m = 0.995$. Table 1 summarizes the classification accuracy of FISF and the other methods. While most nodes have some observed features in uniform-missing settings, $(1 - r_m)$ of nodes do not have observed features at all in structural-missing settings. Therefore, the performance of methods tends to be better in uniform-missing settings than in structural-missing settings. For both missing ways, FISF outperforms the state-of-the-art methods across all the datasets.

## 5.4 LINK PREDICTION RESULTS

Table 2 summarizes the ROC AUC score on link prediction tasks at $r_m = 0.995$. (The AP comparison results are in Appendix F.1.) NIP denotes node-wise inter-channel propagation included in PCFI (Um et al., 2023), which refines an output matrix from channel-wise diffusion. Since NIP is effective in link prediction tasks, we demonstrate the ROC AUC score of FISF and FISF+NIP (FISF followed by NIP). FISF and FISF+NIP achieve state-of-the-art performance in three and four settings, respectively, out of 10 settings. Even in the remaining three settings, FISF+NIP still demonstrates the second-best scores which are comparable with the best scores. That is, FISF and FISF+NIP achieve strong performance across all five datasets regardless of missing ways. As highlighted scores in Table 2 shows, FISF demonstrates its effectiveness on link prediction tasks with missing features.

Table 3: Classification results measured by Micro-F1 score (%). OOM denotes an out-of-memory error.

| Approach | Method | Echocardiogram $(r_m = 2.59\%)$ | ABIDE $(r_m = 52.52\%)$ | Duke Breast Cancer $(r_m = 11.94\%)$ | Diabetes $(r_m = 4.03\%)$ |
|---|---|---|---|---|---|
| Tabular Imputation | GAIN | $68.67 \pm 4.99$ | $89.30 \pm 1.81$ | $76.31 \pm 1.32$ | $53.58 \pm 0.59$ |
| | MIWAE | $69.43 \pm 6.25$ | $64.33 \pm 0.93$ | OOM | OOM |
| | GRAPE | $75.00 \pm 0.81$ | $\mathbf{91.61 \pm 0.89}$ | OOM | OOM |
| | IGRM | $69.33 \pm 8.21$ | $66.38 \pm 1.85$ | OOM | OOM |
| Graph Imputation | GCNMF | $86.00 \pm 2.49$ | $75.05 \pm 2.94$ | $74.86 \pm 1.36$ | $52.17 \pm 0.85$ |
| | FP | $85.67 \pm 4.67$ | $90.79 \pm 1.44$ | $75.38 \pm 2.82$ | $53.03 \pm 0.85$ |
| | PCFI | $86.33 \pm 2.87$ | $90.56 \pm 1.21$ | $75.85 \pm 2.11$ | $52.37 \pm 1.36$ |
| | FISF | $\mathbf{86.67 \pm 2.36}$ | $90.94 \pm 1.45$ | $\mathbf{76.58 \pm 0.62}$ | $\mathbf{53.75 \pm 1.01}$ |

## 5.5 Applicability to Medical Tabular Data

To demonstrate the wide applicability of FISF, we conduct experiments in medical classification using medical tabular datasets, which initially contain missing features. We utilize four medical tabular datasets: Echocardiogram (Asuncion et al., 2007), ABIDE (Di Martino et al., 2014), Duke Breast Cancer (Saha et al., 2018), and Diabetes (Asuncion et al., 2007). In addition to graph imputation methods, since we address imputation on tabular data, we further compare FISF with four imputation methods developed for tabular datasets, including GAIN (Yoon et al., 2018), MIWAE (Mattei & Frellsen, 2019), GRAPE (You et al., 2020), and IGRM (Zhong et al., 2023). For graph imputation methods, we select the three most competitive baselines: GCNMF, FP, and PCFI. We simply construct k-nearest neighbor (kNN) graphs to apply graph imputation methods including our FISF to tabular datasets. The goal of these experiments is to classify each patients, *i.e.*, disease diagnosis.

Table 3 presents the results of medical classification on tabular datasets. As shown in the table, FISF consistently exhibits the best classification performance among graph data imputation methods. Notably, FISF, developed for graph-structure data, also surpasses tabular imputation methods on the Echocardiogram, Duke Breast Cancer, and Diabetes datasets, which do not have predefined connectivity among samples. This indicates the potential for extending graph data imputation to the tabular domain. Furthermore, while MIWAE, GRAPE, and IGRM, which are state-of-the-art tabular imputation methods, suffer from scalability issues, graph imputation methods, including our FISF, operate well across all datasets. Throughout these experiments, we confirm that FISF is effective even in medical classification on tabular datasets initially containing missing values, which are not graph-structured data.

We provide in-depth analyses in Appendix C, including an ablation study, the 'missing not at random' setting, addressing both incomplete features and structure, time complexity, scalability, evidence of little contribution of low-variance channels in downstream tasks, hyperparameter sensitivity, smoothness analysis, and non-uniform-distribution-sampled synthetic features. Additionally, we provide comprehensive discussions, including the justification for synthetic feature injection, in Appendix D.

## 6 Conclusion

In this paper, we identify the low variance problem, which acts as a bottleneck in diffusion-based imputation methods. Based on this important discovery, we propose a novel scheme called Feature Imputation with Synthetic Features (FISF) for graph feature imputation. FISF effectively addresses the problem of low-variance channels by injecting synthetic features, leading to significant performance improvements in both semi-supervised node classification and link prediction tasks. We have verified that FISF consistently demonstrates superiority across various missing rates $r_m$, including low $r_m$. We strongly believe that our work will be widely applied to diverse real-world scenarios that handles graphs with missing features, as our synthetic feature scheme is simple to use and offers significant performance gains. Given the considerable research interest in addressing extremely high rates of missing data across various fields, we anticipate that our FISF will serve as an effective solution for datasets with substantial missingness.

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

## A PROOF OF CONVERGENCE OF DIFFUSION STAGES

Our FISF consists of two diffusion stages: pre-diffusion and DSF. Both stages utilize row stochastic transition matrices for diffusion. We prove the convergence of the two diffusion stages as follows.

**Proposition 1.** *The pre-diffusion transition matrix for the $a$-th channel is defined by*

$$\widetilde{\mathbf{W}}^{(a)} = \begin{bmatrix} \mathbf{I}_{kk} & \mathbf{0}_{ku} \\ \overline{\mathbf{W}}_{uk}^{(a)} & \overline{\mathbf{W}}_{uu}^{(a)} \end{bmatrix},$$

*where $\widetilde{\mathbf{W}}^{(a)}$ is row-stochastic. Using $\widetilde{\mathbf{W}}^{(a)}$, the pre-diffusion in the $a$-th channel is defined by*

$$\tilde{\mathbf{x}}^{(a)}(t) = \widetilde{\mathbf{W}}^{(a)} \tilde{\mathbf{x}}^{(a)}(t-1), \ \ t = 1, \cdots, K;$$

$$\tilde{\mathbf{x}}^{(a)}(0) = \begin{bmatrix} \mathbf{x}_k^{(a)} \\ \mathbf{0}_u \end{bmatrix},$$

*Then, $\lim_{K \to \infty} \tilde{\mathbf{x}}^{(a)}(K)$ converges.*

The proof of Propostion 1 refers to Um et al. (2023). After we establish the convergence of pre-diffusion, we demonstrate that this proof extends to cover the convergence of DSF. To start, we introduce two lemmas.

**Lemma 1.** $\overline{\mathbf{W}}^{(a)}$ *is the row-stochastic matrix calculated by* $\overline{\mathbf{W}}^{(a)} = (\mathbf{D}^{(a)})^{-1}\mathbf{W}^{(a)}$ *where* $\mathbf{D}^{(a)}$ *is a diagonal matrix that has diagonal entities* $\mathbf{D}_{ii}^{(a)} = \sum_j \mathbf{W}_{i,j}$. $\overline{\mathbf{W}}_{uu}^{(a)}$ *is the* $|\hat{\mathbf{x}}_u^{(a)}| \times |\hat{\mathbf{x}}_u^{(a)}|$ *bottom-right submatrix of* $\overline{\mathbf{W}}^{(a)}$ *and let* $\rho(\cdot)$ *denote spectral radius. Then,* $\rho(\overline{\mathbf{W}}_{uu}^{(a)}) < 1$.

*Proof.* Consider $\overline{\mathbf{W}}_{uu0}^{(a)} \in \mathbb{R}^{N \times N}$, where the bottom right submatrix is denoted as $\overline{\mathbf{W}}_{uu}^{(a)}$ and all other elements are zero. That is,

$$\overline{\mathbf{W}}_{uu0}^{(a)} = \begin{bmatrix} \mathbf{0}_{kk} & \mathbf{0}_{ku} \\ \mathbf{0}_{uk} & \overline{\mathbf{W}}_{uu}^{(a)} \end{bmatrix}$$

where $\mathbf{0}_{kk} \in \{0\}^{|\hat{\mathbf{x}}_k^{(a)}| \times |\hat{\mathbf{x}}_k^{(a)}|}$, $\mathbf{0}_{ku} \in \{0\}^{|\hat{\mathbf{x}}_k^{(a)}| \times |\hat{\mathbf{x}}_u^{(a)}|}$, and $\mathbf{0}_{uk} \in \{0\}^{|\hat{\mathbf{x}}_u^{(a)}| \times |\hat{\mathbf{x}}_k^{(a)}|}$. Given that $\overline{\mathbf{W}}^{(a)}$ represents the weighted adjacency matrix of the connected graph $\mathcal{G}$, $\overline{\mathbf{W}}_{uu0}^{(a)} \leq \overline{\mathbf{W}}^{(a)}$ element-wise and $\overline{\mathbf{W}}_{uu0}^{(a)} \neq \overline{\mathbf{W}}^{(a)}$. Furthermore, considering that $\overline{\mathbf{W}}_{uu0}^{(a)} + \overline{\mathbf{W}}^{(a)}$ constitutes the weighted adjacency matrix of a strongly connected graph, we can conclude that $\overline{\mathbf{W}}_{uu0}^{(a)} + \overline{\mathbf{W}}^{(a)}$ is irreducible based on Theorem 2.2.7 in Berman & Plemmons (1994). Consequently, applying Corollary 2.1.5 in Berman & Plemmons (1994), $\rho(\overline{\mathbf{W}}_{uu0}^{(a)}) < \rho(\overline{\mathbf{W}}^{(a)})$. Since the spectral radius of a stochastic matrix is one according to Theorem 2.5.3 in Berman & Plemmons (1994), we have $\rho(\overline{\mathbf{W}}^{(a)}) = 1$. Moreover, since both $\overline{\mathbf{W}}_{uu0}^{(a)}$ and $\overline{\mathbf{W}}_{uu}^{(a)}$ share the same non-zero eigenvalues, it follows that $\rho(\overline{\mathbf{W}}_{uu0}^{(a)}) = \rho(\overline{\mathbf{W}}_{uu}^{(a)})$. Ultimately, this leads to the conclusion that $\rho(\overline{\mathbf{W}}_{uu}^{(a)}) = \rho(\overline{\mathbf{W}}_{uu0}^{(a)}) < \rho(\overline{\mathbf{W}}^{(a)}) = 1$. $\square$

**Lemma 2.** $\mathbf{I}_{uu} - \overline{\mathbf{W}}_{uu}^{(a)}$ *is invertible where* $\mathbf{I}_{uu}$ *is the* $|\hat{\mathbf{x}}_u^{(a)}| \times |\hat{\mathbf{x}}_u^{(a)}|$ *identity matrix.*

*Proof.* Since 1 is not an eigenvalue of $\overline{\mathbf{W}}_{uu}^{(a)}$ by Lemma 1, 0 is not an eigenvlaue of $\mathbf{I}_{uu} - \overline{\mathbf{W}}_{uu}^{(a)}$. Thus $\mathbf{I}_{uu} - \overline{\mathbf{W}}_{uu}^{(a)}$ is invertible. $\square$

We now prove Propostion 1 as follows.

*Proof.* Unfolding the recurrence relation gives us:

$$\hat{\mathbf{x}}^{(a)}(t) = \begin{bmatrix} \hat{\mathbf{x}}_k^{(a)}(t) \\ \hat{\mathbf{x}}_u^{(a)}(t) \end{bmatrix} = \begin{bmatrix} \mathbf{I}_{kk} & \mathbf{0}_{ku} \\ \overline{\mathbf{W}}_{uk}^{(a)} & \overline{\mathbf{W}}_{uu}^{(a)} \end{bmatrix} \begin{bmatrix} \hat{\mathbf{x}}_k^{(a)}(t-1) \\ \hat{\mathbf{x}}_u^{(a)}(t-1) \end{bmatrix} = \begin{bmatrix} \hat{\mathbf{x}}_k^{(a)}(t-1) \\ \overline{\mathbf{W}}_{uk}^{(a)}\hat{\mathbf{x}}_k^{(a)}(t-1) + \overline{\mathbf{W}}_{uu}^{(a)}\hat{\mathbf{x}}_u^{(a)}(t-1) \end{bmatrix}.$$

Since $\hat{\mathbf{x}}_k^{(a)}(t) = \hat{\mathbf{x}}_k^{(a)}(t-1)$ in the first $|\hat{\mathbf{x}}_k^{(a)}|$ rows, it follows that $\hat{\mathbf{x}}_k^{(a)}(K) = \ldots = \hat{\mathbf{x}}_k^{(a)}$. That is, $\hat{\mathbf{x}}_k^{(a)}(K)$ retains the values of $\mathbf{x}_k^{(a)}$. Therefore, $\lim_{K\to\infty} \hat{\mathbf{x}}_k^{(a)}(K)$ converges to $\mathbf{x}_k^{(a)}$.

Now, we focus solely on the convergence of $\lim_{K\to\infty} \hat{\mathbf{x}}_u^{(a)}(K)$. When we unroll the recursion for the last $|\hat{\mathbf{x}}_u^{(a)}|$ rows,

$$
\begin{aligned}
\hat{\mathbf{x}}_u^{(a)}(K) &= \overline{\mathbf{W}}_{uk}^{(a)}\mathbf{x}_k^{(a)} + \overline{\mathbf{W}}_{uu}^{(a)}\hat{\mathbf{x}}_u^{(a)}(K-1) \\
&= \overline{\mathbf{W}}_{uk}^{(a)}\mathbf{x}_k^{(a)} + \overline{\mathbf{W}}_{uu}^{(a)}(\overline{\mathbf{W}}_{uk}^{(a)}\mathbf{x}_k^{(a)} + \overline{\mathbf{W}}_{uu}^{(a)}\hat{\mathbf{x}}_u^{(a)}(K-2)) \\
&= \ldots \\
&= \left(\sum_{t=0}^{K-1}(\overline{\mathbf{W}}_{uu}^{(a)})^t\right)\overline{\mathbf{W}}_{uk}^{(a)}\mathbf{x}_k^{(a)} + (\overline{\mathbf{W}}_{uu}^{(a)})^K\hat{\mathbf{x}}_u^{(a)}(0)
\end{aligned}
$$

By Lemma 1, $\lim_{K\to\infty}(\overline{\mathbf{W}}_{uu}^{(a)})^K = 0$. Therefore, $\lim_{K\to\infty}(\overline{\mathbf{W}}_{uu}^{(a)})^K\hat{\mathbf{x}}_u^{(a)}(0) = 0$, regardless of the initial state for $\hat{\mathbf{x}}_u^{(a)}(0)$. (we replace $\hat{\mathbf{x}}_u^{(a)}(0)$ with a zero column vector for simplicity.) Hence, our focus shifts to $\lim_{K\to\infty}(\sum_{t=0}^{K-1}(\overline{\mathbf{W}}_{uu}^{(a)})^t)\overline{\mathbf{W}}_{uk}^{(a)}\mathbf{x}_k^{(a)}$.

Given that Lemma 1 establishes $\rho(\overline{\mathbf{W}}_{uu}^{(a)}) < 1$, and Lemma 2 affirms the invertibility of $(\mathbf{I}_{uu} - \overline{\mathbf{W}}_{uu}^{(a)})^{-1}$, the geometric series converges as follows

$$
\lim_{K\to\infty}\hat{\mathbf{x}}_u^{(a)}(K) = \lim_{K\to\infty}\left(\sum_{t=0}^{K-1}(\overline{\mathbf{W}}_{uu}^{(a)})^t\right)\overline{\mathbf{W}}_{uk}^{(a)}\mathbf{x}_k^{(a)} = (\mathbf{I}_{uu} - \overline{\mathbf{W}}_{uu}^{(a)})^{-1}\overline{\mathbf{W}}_{uk}^{(a)}\mathbf{x}_k^{(a)}.
$$

In conclusion, the recursion in the pre-diffusion converges. □

In the case of DSF, the DSF transition matrix $\widetilde{\mathbf{M}}^{(b)}$ in Eq. 8 is also row stochastic. The distinction between $\widetilde{\mathbf{W}}^{(a)}$ and $\widetilde{\mathbf{M}}^{(b)}$ lies solely in the number of channels where diffusion is performed and the sizes of each sub-matrix. Therefore, the convergence of the DSF can also be established through the proof of Proposition 1.

# B    PROOF OF THE PROPOSITION IN SEC 4.3

We refer to the proposition in Sec. 4.3 as Proposition 2.

**Proposition 2.** *In pre-diffusion (channel-wise inter-node diffusion (Um et al., 2023)), when all given known features in the $a$-th channel (i.e., elements in $\mathbf{x}_k^{(a)}$) have the same value $c$, $\lim_{t\to\infty}\tilde{\mathbf{x}}^{(a)}(t)$ becomes a vector where entire elements are equal to $c$.*

*Proof.* In accordance with the given assumption, entire elements in $\mathbf{x}_k^{(a)}$ have the value of $c$. Here, we can initialize $\hat{\mathbf{x}}^{(a)}(0)$ with the same values as $c$. According to the proof of Proposition 1, $\lim_{K\to\infty}\hat{\mathbf{x}}_u^{(a)}(K) = (\mathbf{I}_{uu} - \overline{\mathbf{W}}_{uu}^{(a)})^{-1}\overline{\mathbf{W}}_{uk}^{(a)}\mathbf{x}_k^{(a)}$ and $\hat{\mathbf{x}}_k^{(a)}(K) = \mathbf{x}_k^{(a)}$. This means that initializing $\hat{\mathbf{x}}^{(a)}(0)$ with the values of $c$ does not affect the final output, $\lim_{K\to\infty}\hat{\mathbf{x}}^{(a)}(K)$. Formally, pre-diffusion of which steady state is the same as that of Eq. 5 can be expressed as follows:

$$
\begin{aligned}
&\tilde{\mathbf{x}}^{(a)}(t) = \widetilde{\mathbf{W}}^{(a)}\tilde{\mathbf{x}}^{(a)}(t-1), \ \ t = 1, \cdots, K; \\
&\tilde{\mathbf{x}}^{(a)}(0) = \begin{bmatrix} \mathbf{c}_k \\ \mathbf{c}_u \end{bmatrix},
\end{aligned} \tag{10}
$$

where $\mathbf{c}_k$ and $\mathbf{c}_u$ are column vectors with lengths of $|\mathcal{V}_k^{(a)}|$ and $|\mathcal{V}_u^{(a)}|$, respectively, filled only with the value $c$.

Since $\widetilde{\mathbf{W}}^{(a)}$ is row stochastic, $\sum_{j=0}^{K-1} \widetilde{\mathbf{W}}_{i,j}^{(a)} = 1$ for all $i \in \{1, \ldots, N\}$. Therefore, in Eq. 10 , the $i$-th element in $\tilde{\mathbf{x}}^{(a)}(1)$ is calculated as $\sum_{j=0}^{K-1} \widetilde{\mathbf{W}}_{i,j}^{(a)} \cdot c = c \cdot \sum_{j=0}^{K-1} \widetilde{\mathbf{W}}_{i,j}^{(a)} = c$ for all $i \in \{1, \ldots, N\}$. That is, $\tilde{\mathbf{x}}^{(a)}(1)$ is filled only with the value $c$, which is the same as $\tilde{\mathbf{x}}^{(a)}(0)$. Thus, even if this recursion repeats, $\tilde{\mathbf{x}}^{(a)}(t)$ remains the same as $\begin{bmatrix} \mathbf{c}_k \\ \mathbf{c}_u \end{bmatrix}$, which results in $\lim_{t \to \infty} \tilde{\mathbf{x}}^{(a)}(t) = \begin{bmatrix} \mathbf{c}_k \\ \mathbf{c}_u \end{bmatrix}$ where entire elements are equal to $c$. $\qquad\square$

## C  ADDITIONAL EXPERIMENTS

### C.1  ABLATION STUDY

Table 4: Ablation study of FISF. SS node classification denotes semi-supervised node classification. # denotes the number of synthetic features injected into a low-variance channel.  * denotes the optimal hyperparameter at the setting.

| Task | | | SS node classification | Link prediction | |
|---|---|---|---|---|---|
| Dataset | | | CORA | CITESEER | |
| # | $\beta$ | $\gamma$ | ACC | AUC | AP |
| 1 | 1 | 0 | $74.62 \pm 1.78$ | $79.38 \pm 1.81$ | $82.98 \pm 0.86$ |
| 1 | 1 | 100 | $78.50 \pm 1.91$ | $83.63 \pm 1.69$ | $85.42 \pm 1.79$ |
| 1 | 1 | * | $78.52 \pm 1.94$ | $83.46 \pm 1.84$ | $85.32 \pm 1.59$ |
| 1 | * | 100 | $78.78 \pm 1.51$ | $58.67 \pm 13.44$ | $60.27 \pm 14.40$ |
| 2 | * | * | $78.88 \pm 1.91$ | $82.11 \pm 2.43$ | $83.61 \pm 2.50$ |
| 1 | * | * | $\mathbf{79.29 \pm 1.72}$ | $\mathbf{84.12 \pm 1.17}$ | $\mathbf{85.85 \pm 1.38}$ |

We conduct an ablation study to investigate the effectiveness of the elements in FISF. We perform both semi-supervised node classification and link prediction.  For ablation study on semi-supervised node classification, we conduct experiments on Cora under a structural-missing setting with $r_m = 0.995$. For link prediction, we utilize CiteSeer under a structural-missing setting with $r_m = 0.995$. $\beta$ takes a role in spreading synthetic features widely and $\gamma$ implies the ratio of selected low-variance channels to diffuse with synthetic features. Table 4 demonstrates the results of the ablation study. The results show that the performance gain by introducing synthetic features (*i.e.,* $\gamma \neq 0$) is significant. The optimal $\beta$ and the optimal $\gamma$ synergistically enhance the performance, resulting in considerable improvements. The bottom two rows in Table 4 demonstrate that injecting two synthetic features into row-variance channels leads to degradation in performance. This shows the validity of injecting a single synthetic feature into a low-variance channel.

### C.2  MISSING NOT AT RANDOM SETTING

Table 5: Performance in semi-supervised node classification on OGBN-Arxiv at $r_m = 0.995$, measured by accuracy (%).

| Missing setting | LP | GCNMF | GRAFFENE | FP | PCFI | FISF |
|---|---|---|---|---|---|---|
| MNAR-I | $67.56 \pm 0.00$ | $60.73 \pm 0.91$ | $14.60 \pm 4.68$ | $68.63 \pm 0.35$ | $68.24 \pm 0.67$ | $\mathbf{69.02 \pm 0.57}$ |
| MNAR-D | $67.56 \pm 0.00$ | $60.89 \pm 0.52$ | $14.47 \pm 4.54$ | $68.08 \pm 0.41$ | $67.88 \pm 0.29$ | $\mathbf{68.51 \pm 0.25}$ |

Our FISF is a generic method that is effective regardless of missing settings. To further validate the effectiveness of FISF beyond the random missing setting, we conduct additional experiments on 'Missing Not At Random' (MNAR) scenarios.  In MNAR scenarios, the missing probability depends on the unobserved values themselves.  Thus, for the experiments, we establish two MNAR settings: MNAR-I and MNAR-D. In MNAR-I, the missing probability of a feature increases as the feature's value increases; conversely, in MNAR-D, the missing probability decreases as the feature's value increases. For MNAR-I and MNAR-D, we set the missing probability of $x_{i,a}$ to $\max(1, \exp(\frac{x_{i,a}}{(\max(\mathbf{X}) - \min(\mathbf{X}))}))$ and $\max(1, \exp(\frac{-x_{i,a}}{(\max(\mathbf{X}) - \min(\mathbf{X}))}))$, respectively. Table 5 shows

classification accuracy in semi-supervised node classification on the OGBN-Arxiv dataset under MNAR settings. The results reveal that FISF consistently outperforms the baselines across both MNAR settings, thereby demonstrating its effectiveness even in MNAR scenarios.

## C.3 Effectiveness in addressing both incomplete features and structure

Table 6: Performance on semi-supervised node classification tasks at $r_m = 0.995$, measured by accuracy (%).

| $r_m$ | 0.3 | | 0.8 | |
|---|---|---|---|---|
| Dataset | T2-GNN | FISF | T2-GNN | FISF |
| Cora | $84.71 \pm 1.33$ | $\mathbf{87.69 \pm 1.99}$ | $58.03 \pm 1.98$ | $\mathbf{84.75 \pm 1.57}$ |
| CiteSeer | $74.72 \pm 2.96$ | $\mathbf{76.89 \pm 1.08}$ | $54.48 \pm 3.87$ | $\mathbf{75.66 \pm 1.22}$ |
| PubMed | OOM | $\mathbf{85.47 \pm 0.59}$ | OOM | $\mathbf{82.51 \pm 0.47}$ |

Although our FISF cannot reconstruct a missing structure, we verify its effectiveness in addressing a downstream task where both features and structure are incomplete, which T2-GNN (Huo et al., 2023) targets. Moreover, FISF surpasses T2-GNN under the settings specified in Huo et al. (2023). FISF does not make reconstructed features closely resemble their unknown original values, but impute missing features with values that aid in downstream tasks. Thus, FISF is less impacted by missing edges since it can still produce features beneficial for downstream tasks by using remaining edges.

For T2-GNN (Huo et al., 2023), we use the officially released code by the authors and conduct experiments using the label splits provided in huo2023t2. In each label split, nodes are allocated into training, validation, and testing sets with proportions of 60%, 20%, and 20%, respectively. We adhere to the missing settings used in huo2023t2. We apply uniform missing for feature missing, and missing edges are selected at random. For each setting, we report average accuracy with standard deviation across five independent runs.

Table 6 present semi-supervised node classification results under settings where $r_m\%$ of both features and structure are missing, with $r_m$ set to 30% and 80%. The results clearly show that FISF surpasses T2-GNN by a considerable margin across all the settings. Notably, at a higher missing rate of 80%, the advantages of FISF over T2-GNN are especially pronounced. Additionally, FISF shows better scalability compared to T2-GNN, which encounters an out-of-memory error on the PubMed dataset. We leave the development of a diffusion-based imputation method that specifically addresses both feature and structure missing for future research.

## C.4 Complexity Analysis

Table 7: Performance on semi-supervised node classification tasks at $r_m = 0.995$, measured by accuracy (%). FastFISF denotes FISF using FP instead of PCFI for pre-diffusion.

| | | | ***Structural missing*** | | | | |
|---|---|---|---|---|---|---|---|
| Method | Cora | CiteSeer | PubMed | Photo | Computers | OGBN-Arxiv | Average |
| FISF | $79.29 \pm 1.72$ | $69.68 \pm 2.47$ | $76.90 \pm 1.50$ | $88.22 \pm 0.79$ | $79.40 \pm 1.11$ | $69.92 \pm 0.17$ | 77.24 |
| FastFISF | $78.94 \pm 1.92$ | $69.42 \pm 1.44$ | $77.14 \pm 0.94$ | $88.10 \pm 1.38$ | $79.09 \pm 1.42$ | $69.53 \pm 0.21$ | 77.04 |

| | | | ***Uniform missing*** | | | | |
|---|---|---|---|---|---|---|---|
| Method | Cora | CiteSeer | PubMed | Photo | Computers | OGBN-Arxiv | Average |
| FISF | $79.09 \pm 1.73$ | $69.52 \pm 1.81$ | $77.53 \pm 1.28$ | $88.32 \pm 1.37$ | $82.12 \pm 0.51$ | $69.81 \pm 0.16$ | 77.73 |
| FastFISF | $79.29 \pm 1.84$ | $69.39 \pm 1.57$ | $77.41 \pm 1.77$ | $88.03 \pm 1.46$ | $81.70 \pm 0.54$ | $69.45 \pm 0.18$ | 77.55 |

Here we discuss the complexity of FISF which involves two diffusion stages: pre-diffusion and diffusion with synthetic features. FISF takes $O(|\mathcal{E}| + (1 + \gamma F)N^2)$ time under structural-missing settings. Under uniform-missing settings, FISF takes $O(|\mathcal{E}| + (1 + \gamma)FN^2)$ time.

Table 8: Running time of methods. OOM denotes an out-of-memory error.

| Missing way | structural | | uniform | |
|---|---|---|---|---|
| Method | CORA | PUBMED | CORA | PUBMED |
| GCNMF | 10.3s | 19.4s | 9.87s | 28.3s |
| GRAFENNE | 47.9s | 74.7s | 51.1s | 74.0s |
| MEGAE | 1753s | OOM | 1801s | OOM |
| FP | 2.36s | 3.12s | 2.25s | 2.90s |
| PCFI | 2.45s | 3.23s | 11.1s | 34.1s |
| FastFISF | 13.4s | 34.6s | 11.8s | 42.5s |
| FISF | 13.4s | 34.8s | 17.6s | 78.2s |

We observe that the majority of the computation time in FISF is consumed by employing Dijkstra's algorithm to calculate the shortest path distance for each channel. The time complexity of Dijkstra's algorithm is $O(N^2)$. In pre-diffusion under structural missing settings, Dijkstra's algorithm is once utilized since nodes with observed features are equal across all the channels. However, under uniform-missing settings, the time complexity of pre-diffusion increases to $O(N^2F)$, considering the use of Dijkstra's algorithm across all channels.

We can utilize not only channel-wise inter-node diffusion in PCFI but also FP for pre-diffusion. We introduce a variant called FastFISF, which utilizes FP for pre-diffusion, offering efficiency by bypassing the calculation of the shortest path distance. Table 7 demonstrates the results of FastFISF compared to the original FISF on semi-supervised node classification tasks. For channels that are not low-variance channels, features obtained via pre-diffusion are maintained until the end of diffusion with synthetic features. Therefore, since PCFI outperforms FP in terms of performance in downstream tasks, FISF shows slightly better performance than FastFISF in most cases. However, since the performance of FastFISF is comparable to that of FISF, FastFISF can serve as a rapid alternative to FISF without a significant loss in performance.

To address the increasing time complexity in uniform-missing settings, we can employ FastFISF where the time complexity is $O(|\mathcal{E}| + \gamma F N^2)$ regardless of the missing way. Therefore, to address the increasing time complexity of FISF in uniform-missing settings, we can employ FastFISF, accompanied by only a slight performance loss.

Table 8 demonstrates the training time of methods. FP has the lowest training time among the methods. However, FISF brings great performance improvement compared to FP. For instance, in structural-missing setups with $r_m = 0.995$, FISF achieves significant gains in node classification accuracy over FP, showing improvements of 7.43% and 4.94% on Cora and PubMed, respectively. We can further confirm that FastFISF significantly decreases the training time in uniform-missing settings.

### C.5 SCALABILITY OF FISF

Table 9: Performance on semi-supervised node classification tasks at $r_m = 0.995$, measured by accuracy (%).

| Dataset | FP | ScalableFISF | FISF |
|---|---|---|---|
| CORA | 71.86 ± 2.82 | 78.25 ± 1.38 | **79.29 ± 1.72** |
| CITESEER | 58.61 ± 1.74 | 68.52 ± 1.82 | **69.68 ± 2.47** |
| PUBMED | 71.96 ± 3.06 | 74.40 ± 2.64 | **76.90 ± 1.50** |
| PHOTO | 85.42 ± 3.16 | 86.98 ± 1.80 | **88.22 ± 0.79** |
| COMPUTERS | 76.62 ± 1.94 | 78.08 ± 1.18 | **79.40 ± 1.11** |
| OGBN-ARXIV | 68.03 ± 0.52 | 68.55 ± 0.42 | **69.92 ± 0.17** |

In FISF, the bottleneck in terms of computation and memory lies in distance encoding, which requires $O(N^2 \cdot F)$ computation and $O(N^2)$ memory usage. However, the core concept of FISF, adding synthetic features to low-variance channels and diffusing them, isn't confined to specific distance encoding methods, enabling the development of scalable yet effective algorithms with minimal modifications. Here, we introduce a lighter version of FISF named ScalableFISF that utilizes FP instead of the distance encoding. FP decreases a computation complexity to $O(|\mathcal{E}|)$ and is vali-

dated as a scalable algorithm in Rossi et al. (2022) through an experiment on a graph with ~2.5M nodes. Specifically, in ScalableFISF, we utilize FP for pre-diffusion and add synthetic features to low-variance channels. Then, by treating the synthetic features as observed features, we simply reapply FP in these low-variance channels, without distance encoding.

Table 9 demonstrates performance on semi-supervised node classification at $r_m = 0.995$ under structural missing settings, measured in accuracy. The results show that ScalableFISF significantly enhances the performance of FP by addressing the low-variance problem. ScalableFISF exhibits decreases in performance compared to FISF, yet ScalableFISF shows reasonable performance and offers advantages in terms of complexity. Therefore, if PCFI reaches its scalability limit on extremely large graphs with high-dimensional features, ScalableFISF can be a good alternative.

### C.6 Contribution of Low-variance Channels in Downstream Tasks

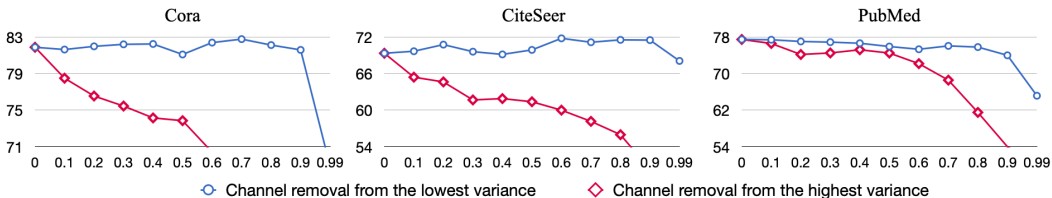

Figure 4: Accuracy (%) on semi-supervised node classification tasks while increasing the proportion of excluded channels from the original feature matrix.

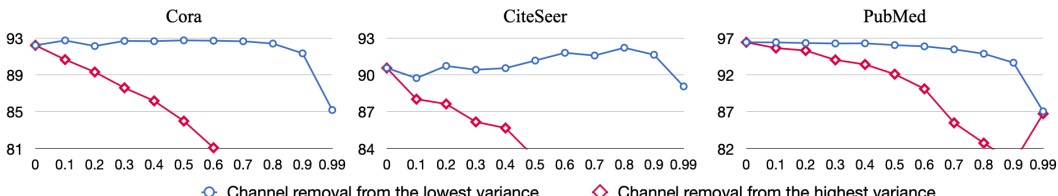

Figure 5: ROC AUC score (%) on link prediction tasks while increasing the proportion of excluded channels from the original feature matrix.

In order to experimentally confirm little contribution of low-variance channels in downstream tasks, we compare performance by excluding partial channels from the original feature matrix using two different ways. The first way (red lines in Figure 4 and Figure 5) is excluding channels in descending order of variance, starting from the highest, based on a fixed proportion. Then, as the second way (blue lines), we exclude channels from the lowest variance in ascending order, *i.e.*, the low-variance channels are removed first.

Figure 4 demonstrates the results on semi-supervised node classification tasks. Since a low-variance channel contains nearly identical values that do not aid in distinguishing nodes, the classification accuracy denoted by blue lines persists despite an increasing removal proportion of low-variance channels. However, cases of channel removal from the highest variance suffer significant performance degradation even with low proportion of channel removal.

As shown in Figure 5, little contribution of low-variance channels is also evident in link prediction tasks. Since identical representations among nodes results in consistent representations across node pairs, low-variance channels also contribute very little to performance in link prediction tasks.

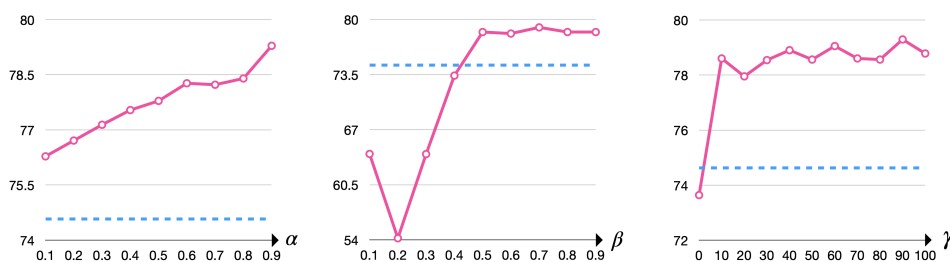

Figure 6: Semi-supervised node classification accuracy with different $\alpha$, $\beta$ and $\gamma$. The blue dashed lines indicate existing state-of-the-art performance.

### C.7   EFFECTS OF HYPERPARAMETERS

We further analyze the effects of FISF hyperparameters, $(\alpha, \beta, \gamma)$, on Cora under structural missing settings with $r_m = 0.995$. Figure 6 shows the accuracy of FISF models with different $\alpha$, $\beta$ and $\gamma$ When varing each hyperparameter, the other hyperparameters are set to their optimal values. Compared to existing state-of-the-art performance of 74.62%, all FISF models consistently exceed it by a considerable margin regardless of the value of $\alpha$. Furthermore, significant performance improvement are observed with a small $\gamma$. A small $\beta$ results in the performance degradation. This is because too small $\beta$ assigns excessive influence to synthetic features, which hinders the spread of known features. This result validates the DSF stage, which enables the wide spread of synthetic features, is properly designed.

### C.8   EFFECTS OF THE MAGNITUDE OF SYNTHETIC FEATURE VALUES

Table 10: Accuracy (%) of FISF for different values of $m$, the scale factor for random noise, on semi-supervised node classification.

| $m$ | 0.01 | 0.1 | 1 (used) | 10 | 100 |
|---|---|---|---|---|---|
| Cora | $76.83 \pm 1.38$ | $78.72 \pm 1.35$ | $79.29 \pm 1.72$ | $79.65 \pm 1.11$ | $71.09 \pm 8.03$ |
| CiteSeer | $68.10 \pm 2.02$ | $68.69 \pm 2.86$ | $69.68 \pm 2.47$ | $68.95 \pm 3.38$ | $66.68 \pm 2.17$ |
| PubMed | $75.09 \pm 2.12$ | $76.78 \pm 1.98$ | $76.90 \pm 1.50$ | $77.28 \pm 0.71$ | $69.19 \pm 12.55$ |
| Photo | $87.95 \pm 1.20$ | $88.49 \pm 1.04$ | $88.22 \pm 0.79$ | $88.01 \pm 1.34$ | $87.75 \pm 1.64$ |
| Computers | $78.86 \pm 0.76$ | $78.93 \pm 1.23$ | $79.40 \pm 1.11$ | $80.01 \pm 0.20$ | $80.16 \pm 0.79$ |
| OGBN-Arxiv | $68.48 \pm 0.17$ | $69.04 \pm 0.38$ | $69.92 \pm 0.17$ | $69.82 \pm 0.15$ | $69.31 \pm 0.18$ |

To confirm the effects of the magnitude of synthetic feature values, we conduct additional experiments by using a scale factor $m$. The values for the synthetic features are scaled by multiplying them by $m$, after being sampled from a uniform distribution on $[0, 1]$. Table 10 shows the results. As shown in the table, $min\{0.1, 1, 10\}$ generally shows similar performance, while there is a performance decrease in the case of $min\{0.01, 100\}$. We believe that the performance drop for $m = 0.01$ is due to the fact that it barely increases the variance of the channel. For $m = 100$, after the imputation process, the low-variance channels with injected synthetic features will be on a different scale compared to other channels without injected synthetic features, which disrupts the learning process of the downstream GNN.

Table 11: Accuracy (%) of FISF for different values of $m$ on semi-supervised node classification, when normalized features are given.

| $m$ | Cora | CiteSeer | PubMed | Photo | Computers | OGBN-Arxiv |
|---|---|---|---|---|---|---|
| 0.1 | $78.68 \pm 1.78$ | $69.42 \pm 2.31$ | $76.93 \pm 1.11$ | $87.55 \pm 1.64$ | $79.24 \pm 0.42$ | $69.56 \pm 0.30$ |
| 1 | $79.03 \pm 1.45$ | $69.50 \pm 2.50$ | $77.16 \pm 1.11$ | $88.12 \pm 1.43$ | $80.23 \pm 0.65$ | $69.88 \pm 0.21$ |
| 10 | $77.99 \pm 1.58$ | $68.50 \pm 2.11$ | $76.14 \pm 2.05$ | $88.21 \pm 0.85$ | $77.61 \pm 1.62$ | $69.75 \pm 0.22$ |

To generalize the sampling distribution against the magnitude of values in the feature channel, node-wise normalization can be a good solution. We apply node-wise L2 normalization to pre-imputed features where synthetic features will be injected. Table 11 shows the results. We can confirm that $m = 1$ produces maintains robust performance across different datasets. These discussions and experimental results demonstrate that the performance is significantly affected when the magnitude of random noise is either too small or too large. They also suggest that node-wise normalization can be a good solution to handle various scales of features effectively.

## C.9    HYPERPARAMETER SEARCH FOR FISF

Table 12: Performance on semi-supervised node classification tasks at $r_m = 0.995$, measured by accuracy (%).

| | | | *Structural missing* | | | | |
|---|---|---|---|---|---|---|---|
| Method | CORA | CITESEER | PUBMED | PHOTO | COMPUTERS | OGBN-ARXIV | Average |
| FISF | $79.29 \pm 1.72$ | $69.68 \pm 2.47$ | $76.90 \pm 1.50$ | $88.22 \pm 0.79$ | $79.40 \pm 1.11$ | $69.92 \pm 0.17$ | 77.24 |
| FISF* | $78.68 \pm 1.72$ | $69.68 \pm 2.47$ | $76.74 \pm 1.84$ | $88.22 \pm 0.79$ | $79.40 \pm 1.11$ | $69.92 \pm 0.17$ | 77.11 |

| | | | *Uniform missing* | | | | |
|---|---|---|---|---|---|---|---|
| Method | CORA | CITESEER | PUBMED | PHOTO | COMPUTERS | OGBN-ARXIV | Average |
| FISF | $79.09 \pm 1.73$ | $69.52 \pm 1.81$ | $77.53 \pm 1.28$ | $88.32 \pm 1.37$ | $82.12 \pm 0.51$ | $69.81 \pm 0.16$ | 77.73 |
| FISF* | $79.09 \pm 1.73$ | $69.52 \pm 1.81$ | $76.89 \pm 2.01$ | $88.32 \pm 1.37$ | $81.56 \pm 0.47$ | $69.81 \pm 0.16$ | 77.53 |

Table 13: Performance on link prediction tasks at $r_m = 0.995$, measured in ROC AUC score (%).

| | | | *Structural missing* | | | |
|---|---|---|---|---|---|---|
| Method | CORA | CITESEER | PUBMED | PHOTO | COMPUTERS | Average |
| FISF | $87.26 \pm 1.44$ | $84.12 \pm 1.17$ | $83.19 \pm 0.78$ | $95.86 \pm 0.21$ | $94.70 \pm 0.30$ | 89.03 |
| FISF* | $86.80 \pm 1.27$ | $84.12 \pm 1.17$ | $82.46 \pm 0.94$ | $95.76 \pm 0.33$ | $94.39 \pm 0.82$ | 88.70 |

| | | | *Uniform missing* | | | |
|---|---|---|---|---|---|---|
| Method | CORA | CITESEER | PUBMED | PHOTO | COMPUTERS | Average |
| FISF | $87.44 \pm 0.80$ | $83.45 \pm 2.53$ | $85.33 \pm 0.47$ | $96.64 \pm 0.18$ | $95.13 \pm 0.35$ | 89.60 |
| FISF* | $87.56 \pm 1.29$ | $81.15 \pm 1.17$ | $82.46 \pm 0.69$ | $95.68 \pm 0.42$ | $94.94 \pm 0.27$ | 88.36 |

Despite the outperforming performance of FISF, conducting a hyperparameter search for FISF with three hyperparameters ($\alpha$, $\beta$, and $\gamma$) can be burdensome in certain situations. However, both $\alpha$ and $\beta$ ($0 < \alpha, \beta < 1$) play a shared role in a base of distance during calculating PC (*i.e.* $\xi_{i,b}^* = \alpha^{\mathbf{S}_{i,b}^*}$ and $\xi_{i,a}^s = \beta^{\mathbf{S}_{i,a}^s}$). Thus we can combine them into one, i.e., $\alpha = \beta$. By doing this, the search complexity can be reduced from $5^3$ to $5^2$ without the performance degradation by setting five search points for each hyperparameter. Table 12 and Table 13 show that the FISF* with the light search does not degrade performance on semi-supervised node classification and link prediction. The version with the light search requires from 20 minutes to 10 hours depending on the datasets, therefore this burden is manageable for practical usage of FISF.

## C.10    SMOOTHNESS ANALYSIS

We generate a synthetic feature in a low-variance channel in order to make features in that channel distinctive across nodes. To investigate smoothness (feature homophily), we compare the smoothness of output features obtained through imputation methods. For this comparison, we employ Dirichlet energy, a representative criterion for measuring smoothness on a graph. As shown in Table 14, FP displays the lowest Dirichlet energy among the imputation methods. In contrast, FISF makes Dirichlet energy of the imputed features similar to that of the original features. Note that our FISF shows the highest Dirichlet energy (distinctiveness) among the methods. Through the outperforming performance of FISF over the existing methods, we can confirm that features with low

Table 14: $\log(E_D)$ of imputed features under a structural-missing setting with $r_m = 0.995$, where $E_D$ is the Dirichlet energy. Original denotes original given features.

| Missing way | Structural | | | Uniform | | |
|---|---|---|---|---|---|---|
| Method ↓ | CORA | CITESEER | PUBMED | CORA | CITESEER | PUBMED |
| Original | 4.36 | 4.49 | 3.11 | 4.36 | 4.49 | 3.11 |
| FP | 1.90 | 1.94 | 0.798 | 1.89 | 1.91 | 0.805 |
| PCFI | 3.14 | 2.59 | 1.49 | 2.52 | 2.64 | 1.43 |
| FISF (Ours) | 3.25 | 2.92 | 4.15 | 2.69 | 2.70 | 4.34 |

dirichlet energy (high feature homophily) does not always ensure good performance in downstream tasks while smoothness is an inductive bias of GNNs.

Table 15: Average cosine similarity of imputed features by FISF, under a structural-missing setting with $r_m = 0.995$.

| Dataset | Inter-class | Intra-class | | | | | | | | Ratio |
|---|---|---|---|---|---|---|---|---|---|---|
| | | class 1 | class 2 | class 3 | class 4 | class 5 | class 6 | class 7 | Average | |
| CORA | 0.760 | 0.858 | 0.902 | 0.902 | 0.844 | 0.691 | 0.826 | 0.870 | 0.842 | 1.11 |
| CITESEER | 0.279 | 0.267 | 0.341 | 0.636 | 0.282 | 0.513 | 0.380 | - | 0.403 | 1.45 |
| PUBMED | 0.871 | 0.893 | 0.936 | 0.880 | - | - | - | - | 0.903 | 1.04 |

Table 16: Average cosine similarity of original features.

| Dataset | Inter-class | Intra-class | | | | | | | | Ratio |
|---|---|---|---|---|---|---|---|---|---|---|
| | | class 1 | class 2 | class 3 | class 4 | class 5 | class 6 | class 7 | Average | |
| CORA | 0.0578 | 0.841 | 0.113 | 0.0896 | 0.683 | 0.0690 | 0.0853 | 0.109 | 0.0883 | 1.53 |
| CITESEER | 0.0470 | 0.655 | 0.0601 | 0.0617 | 0.0650 | 0.762 | 0.0581 | - | 0.0644 | 1.37 |
| PUBMED | 0.0719 | 0.112 | 0.937 | 0.0779 | - | - | - | - | 0.0946 | 1.32 |

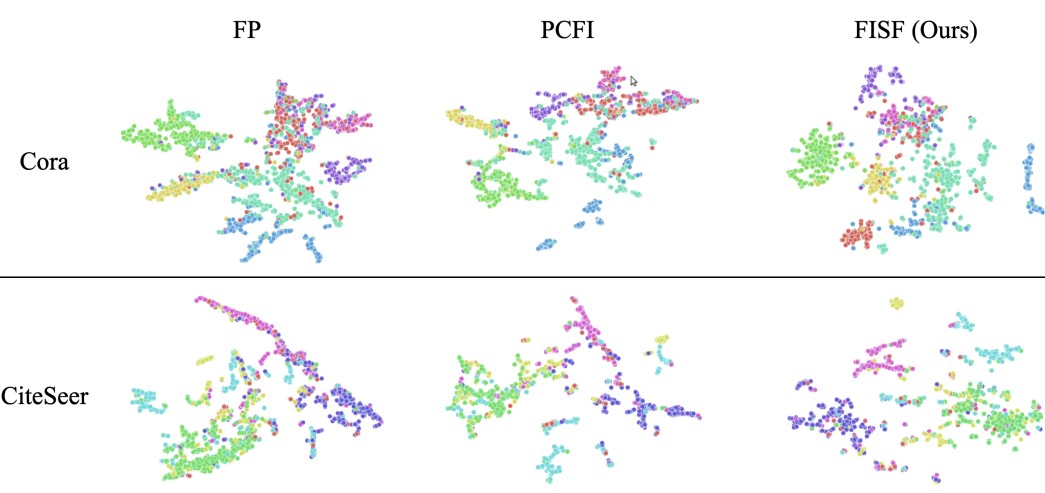

Figure 7: t-SNE plot visualizing imputed features.

To investigate smoothness within classes, we conduct further experiments. Table 15 demonstrates the intra-class cosine similarity calculated from imputed features by FISF. Ratio denotes average similarity/inter-class similarity. If Ratio is greater than 1, inter-class similarity becomes less than the average intra-class similarity, which means the feature is distinctive enough for classification of node features.

Table 16 shows the intra-class cosine similarity calculated from original features. The results indicate that original features also have values of Ratio greater than 1 across the datasets. This means

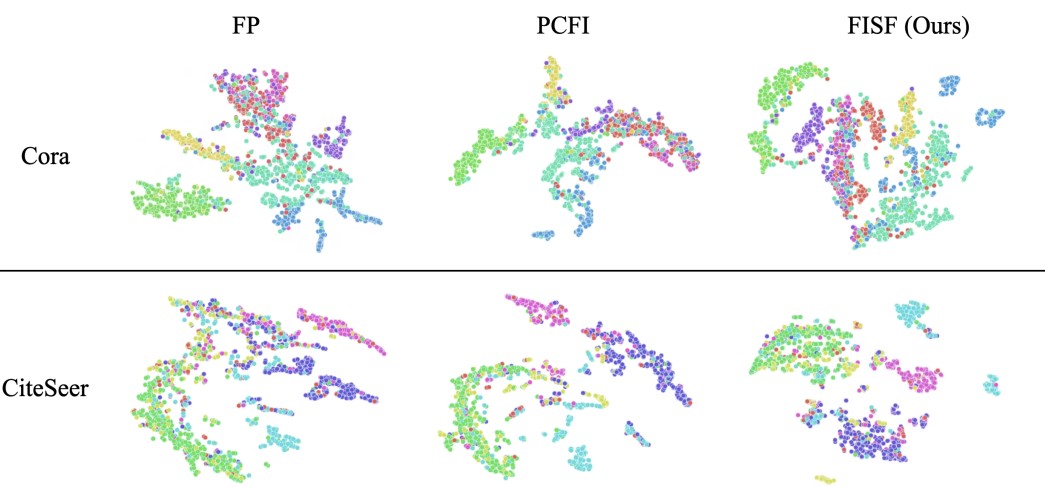

FP                    PCFI                    FISF (Ours)

Cora

CiteSeer

Figure 8: t-SNE plot visualizing deep features in GCN.

that the datasets also originally have higher intra-class feature similarity compared to inter-class feature similarity. Despite the introduction of synthetic features during diffusion, as shown in Table 15, we can observe that imputed features by our scheme consistently maintains higher intra-class feature similarity than inter-class feature similarity.

We also perform qualitative analysis on imputed features and deep features to compare imputation methods. The qualitative analysis is conducted in structural missing settings with $r_m = 0.995$. Figure 7 and Figure 8 demonstrates the t-SNE plots visualizing imputed features and deep features, respectively. FISF provides clearer cluster structures for both imputed features and deep features than the other imputation methods.

## C.11 SYNTHETIC FEATURES SAMPLED FROM A NON-UNIFORM DISTRIBUTION

Table 17: Performance on semi-supervised node classification tasks at $r_m = 0.995$, measured by accuracy (%).

| Dataset | FISF | FISF-L |
|---|---|---|
| CORA | $\mathbf{79.29 \pm 1.72}$ | $78.92 \pm 1.60$ |
| CITESEER | $\mathbf{69.68 \pm 2.47}$ | $69.63 \pm 1.40$ |
| PUBMED | $\mathbf{76.90 \pm 1.50}$ | $76.70 \pm 1.62$ |
| PHOTO | $\mathbf{88.22 \pm 0.79}$ | $88.10 \pm 0.97$ |
| COMPUTERS | $\mathbf{79.40 \pm 1.11}$ | $79.09 \pm 1.14$ |
| OGBN-ARXIV | $\mathbf{69.92 \pm 0.17}$ | $69.03 \pm 0.19$ |

Our FISF samples the value of a synthetic feature from a uniform distribution, because this value only needs to differ from the nearly identical values of observed features within the same channel. Random sampling from a uniform distribution is simple yet effective to achieve this goal. In terms of selecting a node for placing a synthetic feature, we have considered another node sampling scheme that does not rely on a uniform distribution. We attempted to sample the node from a distribution in which the sampling probability varies based on the locations of observed features. We aimed to increase the sampling probability for nodes farther from observed features. However, we empirically observe that sampling the node from this distribution rather degrades performance slightly in downstream tasks, compared to when sampled from a uniform distribution. Table 17 shows the comparison results between the original FISF and FISF-L using the aforementioned node sampling strategy. We believe that this degradation comes from biased selected nodes, which damages the diversity across feature channels in an imputed matrix. Consequently, we sample both a node and a value for a synthetic feature from uniform distributions.

## C.12    Zero Initialization vs Random Initialization

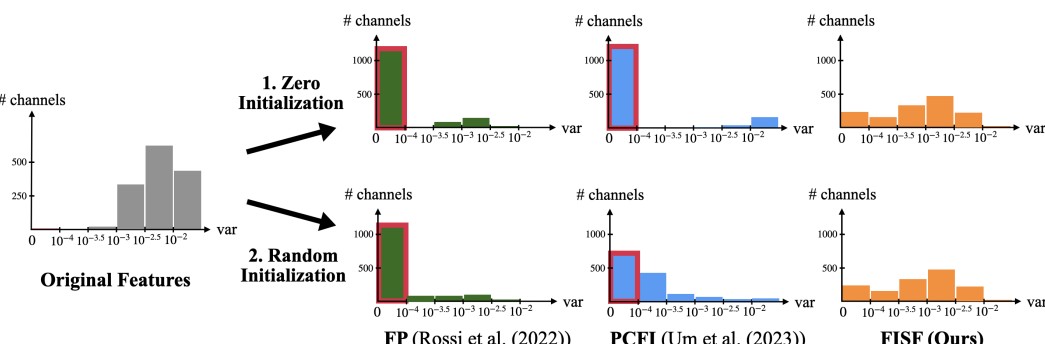

Figure 9: Distributions of variances for each feature channel with zero/random initialization for missing features. Cora dataset with 99.5% missing features is commonly used.

**Do low variance channels occur due to zero initialization use for missing features?** We compare the variance distributions when zero initialization and random initialization are used for missing features. Figure 9 shows that many low-varince channels persist despite random initialization, but there is a slight difference between the distributions despite using the same setting. This is because all the diffusion-based methods approximate the steady state with a sufficiently large hyperparameter $K$, indicating the number of diffusion iteration (e.g., $K = 40$ is used in FP and $K = 100$ is used in PCFI and FISF). However, we have further confirmed that variance distributions becomes identical with very large values (e.g., $K = 1000$) regardless of initialization. Although the final approximated results are not affected by initialization for missing features with a large $K$, careful consideration is needed when determining $K$, depending on the initialization. In conclusion, low-variance channels are not mainly caused by the use of zero initialization for missing features.

## C.13    Statistical Analysis

Table 18: $p$-values comparing our FISF to the runner-up on SSNC, measured across 50 splits of each dataset under structural-missing settings with $r_m = 0.995$. min FISF denotes the worst accuracy among 50 runs.

| Method | Cora | CiteSeer | PubMed | Photo | Computers | OGBN-Arxiv |
|---|---|---|---|---|---|---|
| FISF (ours) | $79.14 \pm 1.32$ | $68.83 \pm 1.95$ | $76.97 \pm 1.44$ | $88.11 \pm 1.21$ | $79.11 \pm 1.01$ | $69.91 \pm 0.22$ |
| min FISF | $75.95$ | $65.43$ | $74.77$ | $86.56$ | $77.09$ | $69.45$ |
| runner-up | $74.35 \pm 1.65$ | $66.01 \pm 2.99$ | $74.53 \pm 2.40$ | $87.44 \pm 1.25$ | $78.72 \pm 1.31$ | $68.78 \pm 0.24$ |
| $p$-value | $1.66 \times 10^{-18}$ | $6.69 \times 10^{-7}$ | $3.13 \times 10^{-8}$ | $2.08 \times 10^{-3}$ | $7.87 \times 10^{-2}$ | $1.96 \times 10^{-28}$ |

We conduct additional experiments to show that our FISF is insensitive to random synthetic feature generation and evaluate the statistical significance of FISF's superior performance. Table 18 shows $p$-values comparing FISF to the runner-up in each setting for the results in semi-supervised node classification tasks under structural missing settings with missing rate ($r_m = 99.5\%$). As shown in the table, the $p$-value indicates the statistical significance of the performance improvement of our FISF over the runner-up. The results demonstrate that our FISF significantly outperforms the runner-up in most cases, with $p$-values much lower than 0.05, suggesting that the performance gains are not due to random chance. Furthermore, even the worst accuracy among 50 runs (min FISF) shows superior or competitive performance compared to the runner-up. This demonstrates that our FISF is robust and insensitive to variations in the random generation of synthetic features, thereby confirming the stability and reliability of our method under extreme missing feature scenarios.

Table 19: $\log(E_D)$ of imputed features on PubMed under structural-missing settings with $r_m = 0.995$.

| $r_m$ | 0.0 | 0.3 | 0.5 | 0.9 | 0.995 | 0.999 |
|---|---|---|---|---|---|---|
| FP | 3.11 | 3.39 | 3.29 | 2.80 | 0.80 | 0.77 |
| PCFI | 3.11 | 3.45 | 3.39 | 3.06 | 1.49 | 2.12 |
| FISF (Ours) | 3.11 | 3.45 | 3.40 | 3.11 | 4.15 | 5.27 |

### C.14 INVESTIGATING THE COUNTERINTUITIVE PERFORMANCE TREND OF FISF UNDER HIGH MISSING RATES

We conduct additional experiments to investigate the underlying reason for the counterintuitive performance trend of FISF at high missing rates, as observed in the PubMed dataset results in Figure 3. Feature homophily, which can be measured by the Dirichlet energy ($E_D$), is a crucial factor for downstream graph neural networks to perform semi-supervised node classification tasks. Hence, we measure the Dirichlet energy ($E_D$) of imputed features. Since this trend is highlighted on PubMed, we perform these experiments on PubMed. Table 19 shows the results. These results indicate that our FISF maintains high Dirichlet energy despite high rates of missing features, while other diffusion-based methods suffer from a severe decrease in Dirichlet energy. The high levels of feature homophily (i.e., Dirichlet energy) stem from synthetic features, which diffuse their values along edges to overcome the low-variance problem.

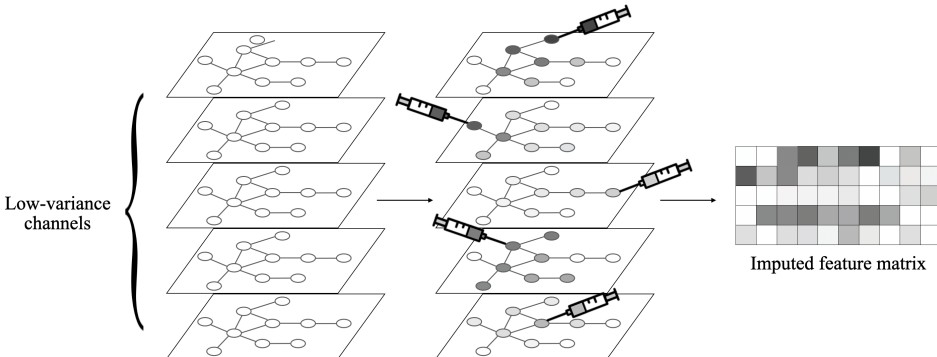

Figure 10: Diffusing a synthetic feature for each low-variance channel results in distinctive imputed features across nodes.

# D DISCUSSIONS

## D.1 JUSTIFICATION FOR SYNTHETIC FEATURE INJECTION

**Conceptual explanation.** In low-variance channels, all missing features are filled with nearly the same values regardless of connectivity, which can not provide any structural information. In contrast, in our scheme, for each low-variance channel, the synthetic feature diffuses its value to its surroundings and creates a local spike centered on the node with the synthetic features. Each node has larger differences in values from the synthetic feature as the distance from the central node increases. If we inject one synthetic feature into each low variance channel, but place it at a different location for each channel. Then the diffused node feature vector containing every low-variance channel feature after diffusion becomes distinctive from those of the other nodes by reflecting the graph structure. Figure 10 illustrates a visualization of the distinctiveness of the diffused feature vector by our scheme.

Table 20: The distribution of channel variances in features imputed by PCFI on PubMed according to $r_m$ under structural-missing settings.

| channel variance | 0.0 | 0.3 | 0.5 | 0.9 | 0.995 | 0.999 |
|---|---|---|---|---|---|---|
| $[10^{-3}, \infty)$ | 13 | 12 | 9 | 1 | 0 | 0 |
| $[10^{-4}, 10^{-3})$ | 467 | 435 | 388 | 126 | 19 | 13 |
| $[10^{-5}, 10^{-4})$ | 20 | 53 | 103 | 373 | 163 | 70 |
| $[0, 10^{-5})$ | 0 | 0 | 0 | 0 | 318 | 417 |

Table 21: The distribution of channel variances in features imputed by FP on PubMed according to $r_m$ under structural-missing settings.

| channel variance | 0.0 | 0.3 | 0.5 | 0.9 | 0.995 | 0.999 |
|---|---|---|---|---|---|---|
| $[10^{-3}, \infty)$ | 13 | 9 | 0 | 0 | 0 | 0 |
| $[10^{-4}, 10^{-3})$ | 467 | 412 | 6 | 58 | 0 | 0 |
| $[10^{-5}, 10^{-4})$ | 20 | 79 | 339 | 439 | 25 | 4 |
| $[0, 10^{-5})$ | 0 | 0 | 155 | 3 | 475 | 496 |

Table 22: The distribution of channel variances in features imputed by FISF on PubMed according to $r_m$ under structural-missing settings.

| channel variance | 0.0 | 0.3 | 0.5 | 0.9 | 0.995 | 0.999 |
|---|---|---|---|---|---|---|
| $[10^{-3}, \infty)$ | 13 | 14 | 12 | 8 | 7 | 0 |
| $[10^{-4}, 10^{-3})$ | 467 | 465 | 414 | 380 | 246 | 222 |
| $[10^{-5}, 10^{-4})$ | 20 | 21 | 74 | 112 | 193 | 205 |
| $[0, 10^{-5})$ | 0 | 0 | 0 | 0 | 54 | 73 |

**Channel variance analysis.** To clarify why randomly sampled values effectively enhance feature distinctiveness, we conduct further experiments that investigate distributions of each channel's variance for varying missing rate ($r_m$). We compare the distributions of the output matrices obtained by our FISF and existing diffusion-based imputation methods. Tables 20, 21 and 22 demonstrate the results. As shown in Tables 20 and 21, the number of low-variance channels in outputs produced by existing diffusion-based imputation methods substantially increases as $r_m$ increases. This implies a decrease in the distinctiveness of imputed features since all features within a low-variance channel have nearly the same values. Unlike these methods, as shown in Table 22, we can confirm that FISF effectively alleviates the occurrence of low-variance channels, indicating significantly higher feature distinctiveness compared to existing methods.

## D.2 Low Variance Problem vs Over-Smoothing Problem

To clarify the distinction between the low variance problem and the over-smoothing problem (Keriven, 2022), we emphasize a fundamental difference between the two issues from the perspective of self-loops. Diffusion-based imputation methods (Rossi et al., 2022; Um et al., 2023) and typical GNNs share a message passing framework to update features using aggregation steps. However, during the aggregation steps of diffusion-based imputation methods, all observed features have self-loops with a weight of 1, while these observed features do not aggregate features from neighboring nodes (*i.e.*, do not consider graph structures). The purpose of this aggregation rule is to preserve the observed features despite multiple aggregation steps, while updating the values of missing features. Due to this different aggregation rule only for observed features, the steady state of overall imputed features is determined by the values of observed features (as shown in Appendix A). We mathematically demonstrate that the cause of low variance channels lies in the situation where the values of observed features within a specific channel are identical (as shown in Appendix B). In a nutshell, the low-variance problem arises from identical values of observed features within a specific channel.

In contrast, typical GNNs that suffer from the over-smoothing problem have consistent update rules, including the weights of self-loops, across nodes and features. All nodes update their features by aggregating features from neighboring nodes. The cause of the over-smoothing problem is proven to be the excessive number of GNN layers (Keriven, 2022; Oono & Suzuki, 2019; Luan et al., 2019). The key point in this proof is that the eigenvalues of the graph Laplacian, which is the weighted matrix used in GNN layers for message passing, fall between 0 and 1. In contrast, although the weighted matrix used in diffusion-based imputation also has eigenvalues between 0 and 1, the reason why the large number of layers does not lead to the over-smoothing problem is due to the aforementioned unique aggregation rule regarding self-loops.

There are two main approaches designed to address the over-smoothing problem. The first is a self-loop-based approach, including APPNP (Gasteiger et al., 2018) and GDC (Klicpera et al., 2019), which adds self-loops with certain weights to all nodes to prevent excessive smoothing. We compare this approach with our FISF by applying an APPNP-style diffusion rule, as shown in Table 6 of the general response PDF. As illustrated in the table, our FISF consistently outperforms the APPNP-style imputation by significant margins across various datasets under structural missing settings with a missing rate of 0.995. The second approach is concatenation-based, where multi-scale features aggregated from neighbors at different hops are concatenated (Luan et al., 2019; Wang et al., 2020). However, since imputation requires output with the same dimension as the original features, the concatenation-based approach developed to address the over-smoothing problem cannot be applied to imputation.

## D.3 Why not Use graph Positional/Structural Encoding?

The key distinction of our FISF approach from graph positional/structural encoding is that FISF allows for the integration of feature information and structural information within low-variance channels. In FISF, although a synthetic feature with randomly sampled values is injected into a low-variance channel, the channel still contains observed features with nearly identical values. Since FISF preserves all observed features during its diffusion process, the output of FISF retains this feature information, reflecting the nearly identical values within the channel. Simultaneously, the injected synthetic feature with a distinct value makes its surrounding nodes similar to its own value,

thereby encoding structural information. After the final diffusion stage, the low-variance channels in the output will contain both nearly identical observed feature values and feature values similar to the synthetic feature, corresponding to feature information and structural information, respectively. Thus, FISF can naturally integrate both feature and structural information within low-variance channels.

Table 23: Performance in semi-supervised node classification on various datasets, measured by accuracy (%).

| Method | Cora | CiteSeer | PubMed | Photo | Computers |
|---|---|---|---|---|---|
| node2vec | $76.67 \pm 1.48$ | $64.00 \pm 1.66$ | $69.50 \pm 4.09$ | $87.77 \pm 1.42$ | $78.98 \pm 1.55$ |
| Preliminary diffusion + node2vec | $77.20 \pm 1.38$ | $66.78 \pm 1.62$ | $70.19 \pm 4.35$ | $87.81 \pm 1.74$ | $79.25 \pm 0.94$ |
| FISF (ours) | $\mathbf{79.29 \pm 1.72}$ | $\mathbf{69.98 \pm 2.47}$ | $\mathbf{76.90 \pm 1.50}$ | $\mathbf{88.20 \pm 0.79}$ | $\mathbf{79.40 \pm 1.11}$ |

We compare our FISF with the case where a positional/structural encoding vector is used as complementary values. We employ node2vec (Grover & Leskovec, 2016), a representative structural encoding method. The table below presents the accuracy in semi-supervised node classification under structural-missing settings with a missing rate of 99.5%, where "node2vec" denotes the case where node2vec is used alone as input, and "Preliminary diffusion + node2vec" refers to the case where node2vec is used as complementary values. As shown in the table, our FISF consistently outperforms both cases using positional/structural encoding vectors across datasets. These performance gains stem from FISF's ability to integrate feature information and structural information within low-variance channels.

# E    EXPERIMENTAL DETAILS

## E.1    DATASET DETAILS

Table 24 summarizes the dataset statistics. All the datasets used in this paper are provided in Pytorch Geometric. All the datasets used in our work, including the Cora, CiteSeer, PubMed, Photo, Computers, and OGBN-arxiv, are MIT-licensed. In the citation networks, nodes and edges represent documents and citation links, respectively. In the case of recommendation networks, nodes represent goods and an edge connects two nodes only when the nodes (*i.e.*, products) are frequently bought together. Following Rossi et al. (2022) and Um et al. (2023), we conduct all experiments on the largest connected graph of each dataset. FISF can also handle disconnected graphs by working on each connected graph.

Table 24: Dataset statistics.

| Dataset | #Nodes | #Edges | #Features | #Classes |
|---|---|---|---|---|
| CORA | 2,485 | 5,069 | 1,433 | 7 |
| CITESEER | 2,120 | 3,679 | 3,703 | 6 |
| PUBMED | 19,717 | 44,324 | 500 | 3 |
| PHOTO | 7,487 | 119,043 | 745 | 8 |
| COMPUTERS | 13,381 | 245,778 | 767 | 10 |
| OGBN-ARXIV | 169,343 | 1,166,243 | 128 | 40 |

## E.2    IMPLEMENTATION DETAILS

We conduct all the experiments on a single NVIDIA GeForce RTX 2080 Ti GPU and an Intel Core I5-6600 CPU at 3.30 Hz. All models are implemented in Pytorch (Paszke et al., 2019) and Pytorch Geometric (Fey & Lenssen, 2019).

**Semi-supervised node classification.** We randomly create 5 different training/validation/test node splits for each dataset except for OGBN-Arxiv. (The node split of OGBN-Arxiv is fixed according to published years of papers (*i.e.,* nodes).) Following the splits in Klicpera et al. (2019), we assign 20 nodes per class as training nodes. Subsequently, the number of validation nodes is adjusted to ensure that when combined with the training nodes, it totals $1,500$. For test nodes, we include all nodes except those designated as training or validation nodes.

Table 25: Statistics of medical tabular datasets.

| Dataset | $N$ | $F$ | $F_{num}$ | $F_{cat}$ | $C$ | $r_m$ |
|---|---|---|---|---|---|---|
| Echocardiogram | 74 | 12 | 3 | 9 | 2 | 2.59% |
| Duke Breast Cancer | 907 | 93 | 34 | 59 | 2 | 11.94% |
| ABIDE | 1112 | 104 | 85 | 19 | 2 | 52.52% |
| Diabetes | 10177 | 47 | 11 | 36 | 3 | 4.03% |

Vanilla GCN models for imputation methods (MEGAE (Gao et al., 2023), FP (Rossi et al., 2022), PCFI (Um et al., 2023), and our FISF) and GCNMF models are trained as follows. We utilize Adam optimizer (Kingma & Ba, 2014) and set the maximum number of epochs to $10,000$. We use an early stopping strategy based on validation accuracy, with a patience of 200 epochs. We apply dropout (Srivastava et al., 2014) with the drop probability $p$. $p$ and learning rates in all experiments are searched in $\{0, 0.25, 0.5\}$ and $\{0.01, 0.005, 0.001, 0.0001\}$, respectively, using grid search on validation sets. We train GRAFENNE models by following the training details specified in Gupta et al. (2023).

For all the baselines, we follow all the hyperparameters specified in the original papers or codes. If hyperparameters (specifically, hidden dimension and the number of layers) for a specific dataset are not clarified in the papers, we perform a hyperparameter search using a grid search approach. The search ranges of hidden dimension and the number of layers are $\{16, 32, 64, 128, 256\}$ and $\{2, 3\}$, respectively.

**Link prediction.** For GCNMF and GAE used as downstream models for imputation methods, we train all the models with Adam optimizer for 200 iterations. We apply dropout (Srivastava et al., 2014) with the drop probability $p$. Through grid search on the validation sets, $p$ and learning rates in all experiments are searched within $\{0, 0.25, 0.5\}$ and $\{0.1, 0.01, 0.005, 0.001, 0.0001\}$, respectively. We randomly create 5 different training/validation/test edge splits for each dataset. For each split, as the splits in Kipf & Welling (2016b), we assign $10\%$ edges for the training set, $5\%$ edges for the validation set, and $85\%$ edges for the test set.

For GAE models for the imputation methods, we commonly train the models as follows. We use Adam optimizer and set the number of epochs to 200. Learning rates are searched from $\{0.01, 0.005, 0.001, 0.0001\}$ by grid search on validation sets. Following Kipf & Welling (2016b), Taguchi et al. (2021), and Um et al. (2023), we leverage GAE models with 32-dimensional hidden layer and 16-dimensional latent variables.

**Medical Classification.** We create five random splits for training, validation, and testing, with proportions of 10%, 10%, and 80%, respectively. The classification performance is then measured by calculating the average Micro-F1 score across these five splits. We utilize MLP classifiers on the feature matrices imputed by tabular imputation methods to perform classification. For the MLP classifiers, we set the number of layers and the hidden dimension to 2 and 64, respectively. Table 25 presents the statistics of the medical tabular datasets used in this paper. $N$ refers to the number of samples, while $F$ indicates the number of features. The numerical and categorical features are represented by $F_{num}$ and $F_{cat}$, respectively. The numerical features are scaled to a fixed range of 0 to 1, and categorical features are encoded using one-hot encoding. $C$ denotes the number of classes, and $r_m$ indicates the missing feature rate in each dataset. The value of $k$ in kNN graph construction for graph imputation methods is selected from $\{1,3,5,10\}$ based on the validation set.

**FISF implementation.** For semi-supervised node classification tasks, we set the number of layers and learning rates to $64$ and $0.005$, respectively. For link prediction tasks on Cora, CiteSeer, and PubMed, we set learning rates to $0.01$. We set learning rates to $0.001$ for Photo and Computers. In all experiments, we fix $K$ to 100 and dropout is applied with $p = 0.5$. In the case of experiments on OGBN-Arxiv, following FP (Rossi et al., 2022) and PCFI (Um et al., 2023), we leverage GCN layers with skip connections (Xu et al., 2018) and set the hidden dimension to 256. Hyperparamters ($\alpha$, $\beta$, and $\gamma$) of FISF used in experiments are summarized in Table 26 and Table 27. We will release the code upon publication.

**Implementation of baselines.** For LP, we use codes implemented in Pytorch Geometric (Fey & Lenssen, 2019). The hyperparameter $\alpha$ of LP is searched from $\{0.95, 0.9, 0.8, 0.7, \ldots, 0.1\}$. For

Table 26: FISF hyperparameters used in experiments on semi-supervised node classification tasks.

| Missing way | Structural missing | | | | | | | | | | | | | | |
|---|---|---|---|---|---|---|---|---|---|---|---|---|---|---|---|
| $r_m$ | 0.3 | | | 0.5 | | | 0.9 | | | 0.995 | | | 0.999 | | |
| Datasets | $\alpha$ | $\beta$ | $\gamma$ | $\alpha$ | $\beta$ | $\gamma$ | $\alpha$ | $\beta$ | $\gamma$ | $\alpha$ | $\beta$ | $\gamma$ | $\alpha$ | $\beta$ | $\gamma$ |
| CORA | 0.7 | 0.9 | 10 | 0.7 | 0.9 | 50 | 0.9 | 0.7 | 90 | 0.9 | 0.7 | 90 | 0.9 | 0.9 | 90 |
| CITESEER | 0.9 | 0.7 | 90 | 0.7 | 0.7 | 30 | 0.9 | 0.5 | 50 | 0.9 | 0.9 | 90 | 0.9 | 0.9 | 90 |
| PUBMED | 0.9 | 0.9 | 10 | 0.9 | 0.7 | 70 | 0.9 | 0.5 | 10 | 0.9 | 0.5 | 90 | 0.9 | 0.5 | 90 |
| PHOTO | 0.5 | 0.9 | 10 | 0.5 | 0.7 | 90 | 0.1 | 0.9 | 70 | 0.1 | 0.1 | 70 | 0.1 | 0.1 | 50 |
| COMPUTERS | 0.3 | 0.9 | 10 | 0.1 | 0.1 | 90 | 0.1 | 0.7 | 50 | 0.1 | 0.1 | 50 | 0.1 | 0.1 | 90 |
| OGBN-ARXIV | 0.3 | 0.3 | 10 | 0.3 | 0.3 | 10 | 0.1 | 0.3 | 30 | 0.1 | 0.1 | 90 | 0.1 | 0.1 | 70 |

| Missing way | Uniform missing | | | | | | | | | | | | | | |
|---|---|---|---|---|---|---|---|---|---|---|---|---|---|---|---|
| $r_m$ | 0.3 | | | 0.5 | | | 0.9 | | | 0.995 | | | 0.999 | | |
| Datasets | $\alpha$ | $\beta$ | $\gamma$ | $\alpha$ | $\beta$ | $\gamma$ | $\alpha$ | $\beta$ | $\gamma$ | $\alpha$ | $\beta$ | $\gamma$ | $\alpha$ | $\beta$ | $\gamma$ |
| CORA | 0.9 | 0.9 | 10 | 0.9 | 0.7 | 30 | 0.7 | 0.9 | 30 | 0.9 | 0.7 | 70 | 0.7 | 0.7 | 70 |
| CITESEER | 0.1 | 0.3 | 50 | 0.1 | 0.3 | 70 | 0.9 | 0.5 | 70 | 0.9 | 0.9 | 30 | 0.7 | 0.7 | 90 |
| PUBMED | 0.3 | 0.1 | 10 | 0.3 | 0.1 | 30 | 0.9 | 0.5 | 50 | 0.9 | 0.5 | 50 | 0.9 | 0.5 | 90 |
| PHOTO | 0.3 | 0.3 | 53 | 0.3 | 0.3 | 50 | 0.1 | 0.3 | 70 | 0.3 | 0.1 | 30 | 0.1 | 0.5 | 90 |
| COMPUTERS | 0.5 | 0.5 | 10 | 0.5 | 0.5 | 10 | 0.1 | 0.3 | 10 | 0.1 | 0.5 | 50 | 0.1 | 0.5 | 50 |
| OGBN-ARXIV | 0.3 | 0.1 | 10 | 0.3 | 0.1 | 30 | 0.9 | 0.3 | 30 | 0.1 | 0.1 | 90 | 0.1 | 0.1 | 10 |

Table 27: FISF hyperparameters used in experiments on link prediction tasks.

| Missing way | Structural missing | | | Uniform missing | | |
|---|---|---|---|---|---|---|
| $r_m$ | 0.995 | | | 0.995 | | |
| Datasets | $\alpha$ | $\beta$ | $\gamma$ | $\alpha$ | $\beta$ | $\gamma$ |
| CORA | 0.5 | 0.9 | 90 | 0.3 | 0.9 | 10 |
| CITESEER | 0.9 | 0.9 | 90 | 0.1 | 0.7 | 10 |
| PUBMED | 0.1 | 0.3 | 70 | 0.1 | 0.5 | 90 |
| COMPUTERS | 0.1 | 0.9 | 10 | 0.1 | 0.9 | 70 |
| PHOTO | 0.1 | 0.7 | 10 | 0.1 | 0.7 | 10 |

Table 28: URL links for baselines.

| Baseline | URL link |
|---|---|
| GCNMF | https://github.com/marblet/GCNmf |
| GRAFENNE | https://github.com/data-iitd/Grafenne |
| MEGAE | https://github.com/zqgao22/max-entropy-gae |
| FP | https://github.com/twitter-research/feature-propagation |
| PCFI | https://github.com/daehoum1/pcfi |

the baselines except for LP, we use code released by the authors of papers. The URL links for the baselines are given in Table 28. While the codes for FP and PCFI are licensed under Apache-2.0, and the codes for GCNMF and MEGAE are licensed under MIT, the code for GRAFENNE has no public declaration of license.

# F  ADDITIONAL EXPERIMENTAL RESULTS

## F.1  AP RESULTS ON LINK PREDICTION

Table 29: Performance on link prediction tasks at $r_m = 0.995$, measured by AP (%). Standard deviation errors are given. The best result is highlighted in bold and underlined, while the second-best result is highlighted only in bold. OOM denotes an out-of-memory error.

| | | | *Structural missing* | | |
|---|---|---|---|---|---|
| Method | CORA | CITESEER | PUBMED | PHOTO | COMPUTERS |
| Full features | $92.62 \pm 1.13$ | $91.60 \pm 1.44$ | $96.59 \pm 0.32$ | $95.24 \pm 0.39$ | $93.77 \pm 0.61$ |
| GCNMF | $70.20 \pm 0.80$ | $69.19 \pm 1.78$ | $86.20 \pm 0.32$ | $80.58 \pm 0.28$ | $83.34 \pm 0.17$ |
| GRAFENNE | $64.70 \pm 3.76$ | $72.08 \pm 9.71$ | $70.43 \pm 3.74$ | $64.78 \pm 0.84$ | $66.56 \pm 1.14$ |
| MEGAE | $69.78 \pm 0.78$ | $70.85 \pm 2.92$ | OOM | $86.46 \pm 1.65$ | $86.12 \pm 1.13$ |
| FP | $86.40 \pm 1.26$ | $82.61 \pm 1.96$ | $83.98 \pm 0.79$ | $93.74 \pm 0.57$ | $91.50 \pm 0.56$ |
| PCFI | $88.63 \pm 0.90$ | $82.98 \pm 0.86$ | $87.07 \pm 0.42$ | $\mathbf{96.31 \pm 0.25}$ | $94.58 \pm 0.37$ |
| FISF | $\mathbf{88.81 \pm 1.35}$ | $\underline{\mathbf{85.85 \pm 1.38}}$ | $\underline{\mathbf{87.55 \pm 0.35}}$ | $95.33 \pm 0.22$ | $\mathbf{94.71 \pm 0.26}$ |
| FISF+NIP | $\underline{\mathbf{89.35 \pm 1.24}}$ | $\mathbf{85.25 \pm 1.85}$ | $\underline{\mathbf{87.62 \pm 0.12}}$ | $\mathbf{95.95 \pm 0.18}$ | $\underline{\mathbf{95.41 \pm 0.33}}$ |

| | | | *Uniform missing* | | |
|---|---|---|---|---|---|
| Method | CORA | CITESEER | PUBMED | PHOTO | COMPUTERS |
| Full features | $92.62 \pm 1.13$ | $91.60 \pm 1.44$ | $96.59 \pm 0.32$ | $95.24 \pm 0.39$ | $93.77 \pm 0.61$ |
| GCNMF | $64.21 \pm 2.01$ | $65.06 \pm 2.67$ | $82.64 \pm 2.17$ | $80.61 \pm 0.20$ | $83.38 \pm 0.12$ |
| GRAFENNE | $75.04 \pm 13.33$ | $71.39 \pm 9.71$ | $73.56 \pm 5.77$ | $68.36 \pm 7.71$ | $69.79 \pm 5.81$ |
| MEGAE | $67.98 \pm 1.85$ | $63.67 \pm 2.89$ | OOM | $83.22 \pm 1.48$ | $85.11 \pm 2.00$ |
| FP | $88.67 \pm 1.26$ | $85.39 \pm 1.89$ | $82.99 \pm 2.14$ | $95.51 \pm 0.19$ | $94.06 \pm 0.27$ |
| PCFI | $89.13 \pm 1.06$ | $\underline{\mathbf{85.47 \pm 1.82}}$ | $88.20 \pm 0.38$ | $\underline{\mathbf{96.87 \pm 0.20}}$ | $\mathbf{95.55 \pm 0.32}$ |
| FISF | $\mathbf{89.16 \pm 0.77}$ | $\mathbf{85.17 \pm 2.00}$ | $\underline{\mathbf{88.73 \pm 0.36}}$ | $96.27 \pm 0.23$ | $95.12 \pm 0.32$ |
| FISF+NIP | $\underline{\mathbf{89.23 \pm 0.89}}$ | $84.73 \pm 2.00$ | $\mathbf{88.72 \pm 0.36}$ | $\mathbf{96.32 \pm 0.26}$ | $\underline{\mathbf{96.12 \pm 0.30}}$ |

### F.2 DISTRIBUTIONS OF FEATURE VARIANCES

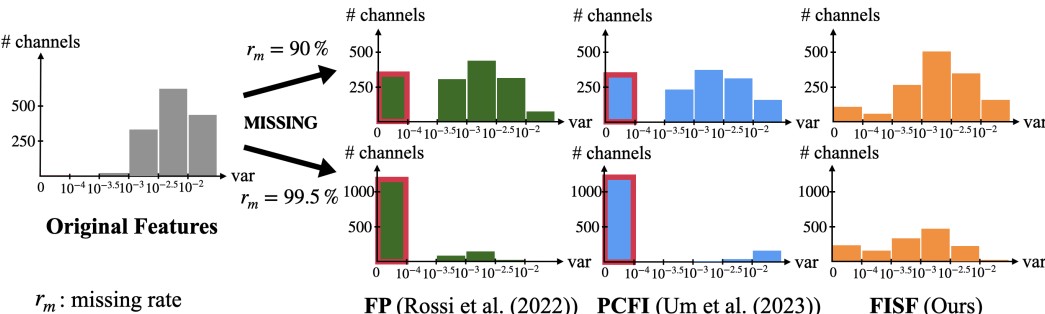

Figure 11: Distributions of variances for each feature channel on Cora dataset with 90%/99.5% missing features. FP and PCFI generates output matrices with many low-variance channels outlined in red, whereas FISF resolves the issue of low-variance channels.

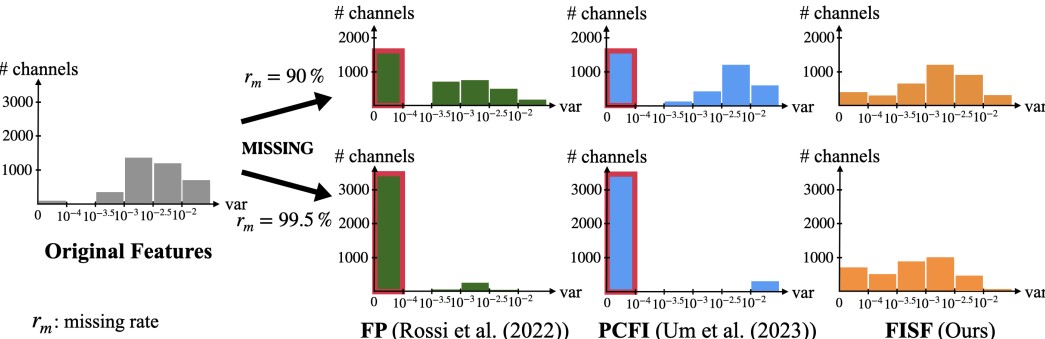

Figure 12: Distributions of variances for each feature channel on CiteSeer dataset with 90%/99.5% missing features. FP and PCFI generates output matrices with many low-variance channels outlined in red, whereas FISF resolves the issue of low-variance channels.

