# OpenReview forum: "Low Variance: A Bottleneck in Diffusion-Based Graph Imputation"
_ICLR.cc/2025/Conference — ICLR 2025 Conference Withdrawn Submission_

### Official Review · Reviewer_Svxx · 2024-11-04

**Soundness:** 2
**Presentation:** 3
**Contribution:** 1
**Rating:** 3
**Confidence:** 4

**Summary:**

The authors found the diffusion methods on graphs to impute missing data reinforce a low variance problem in feature channels, hindering the model performance. They proposed to inject random noise on such channels and re-diffuse on synthetic labels.

**Strengths:**

The authors identified a low variance issue exacerbated during diffusion to fill in the missing values. Clearly such channels do not provide much information in many downstream tasks. They proposed to inject random noise and re-diffuse using a very similary method to PCFI, with a hyperparameter to allow the synthetic data to have wider range influence. The paper is easy to follow.

**Weaknesses:**

My concerns are mostly on the novelty of the paper.

The paper method description, analysis, and experimentation follow PCFI. The authors pointed out that low variance channels are causing issues in downstream task learning. Injecting random noise and then letting the feature diffuse from the “noisy nodes” does increase the variance, help distinguish the nodes and allow some structural information encoded in the process. But the process does not seem to be much different from PCFI, the methods described in this feel more like an implementation variation/details in PCFI.

Additionally, there has been published work to address such issues, as the authors reviewed in 2.2, positional encoding etc.

**Questions:**

The authors argued in D3 that their method is better than positional encoding: in table 20 that under 99.5% missing rate, FIST outperforms positional encoding (node2vec). Can we get some sensitivity analysis on the other missing rate and different positional encoding techniques? Also, some of the experiment details are lacking.

---

### Official Review · Reviewer_A4gM · 2024-11-04

**Soundness:** 2
**Presentation:** 2
**Contribution:** 2
**Rating:** 3
**Confidence:** 3

**Summary:**

In this paper, they address learning tasks on graphs with missing features. Traditional graph diffusion-based imputation methods often yield low-variance channels with nearly identical values, which contribute minimally to graph learning performance. To counteract this, they introduce synthetic features that reduce low-variance production in diffusion-based imputation, thereby enhancing feature diversity. They provide both empirical and theoretical validation of the low-variance issue.

**Strengths:**

1. The proposed method demonstrates promising results across multiple datasets, including a real-world application, highlighting its practical effectiveness and versatility.
2. The method is supported by theoretical analysis, which strengthens the validity of the approach.
3. The application of diffusion-based techniques on graphs is intriguing

**Weaknesses:**

1. The motivation for this study requires further clarification, particularly in establishing a clear connection between missing features and their impact on graph learning performance. The logical link between the presence of missing features and the degradation in model performance is not thoroughly articulated.

2. The problem setting requires further clarification. The term “missing features” is too broad, as it could refer to missing graph structure or node features, each posing distinct challenges. It’s important to specify the type of missing data being addressed and to clearly illustrate the characteristics and implications of different types of feature missingness. A more precise explanation would help readers understand the unique challenges of the specific missing-feature scenario considered in this paper and how it influences the choice of methods.

3.The scalability of the proposed method is not thoroughly discussed, particularly concerning large-scale graphs or graphs with extremely high missing feature rates.

**Questions:**

same as in weakness

---

### Official Review · Reviewer_bu2p · 2024-11-04

**Soundness:** 3
**Presentation:** 3
**Contribution:** 2
**Rating:** 5
**Confidence:** 4

**Summary:**

This paper proposes a novel methodology for solving graph machine learning tasks with missing input node features. The main idea consists of three steps: 1) pre-diffusion; 2) identify low-variance feature channel for synthetic feature injection; 3) post injection diffusion. Experimental results demonstrate empirically the effectiveness of the proposed method.

**Strengths:**

1) The paper investigates an under-explored but practically important setting in graph machine learning.

2) The technical framework is intuitive and simple to implement. It is also presented clearly with the necessarily detail.

3) Experimental results and analyses in Appendix are comprehensive in providing evidence the proposed method works well in practice.

**Weaknesses:**

1) The proposed method, while intuitive, lacks sufficient theoretical justification. It is not entirely clear why injecting random features and re-run diffusion would help, apart from mechanistically forcing features not to converge to uniform values. It would be good if the authors can provide more theoretical investigation of the proposed method, perhaps from the viewpoint of expressivity or spectral analysis. At the moment, the theoretical contribution appears limited in my view.

2) Some modelling choices seem ad-hoc and need more justification and validation (see Questions below).

3) Empirical experiments seem to only focus on datasets where reasonably high homophily in node labels. This somewhat limits the understanding of effectiveness of the proposed method. It would be good to see the method tested against baselines in low homophily settings.

**Questions:**

1) In my view, the notion of low-variance channels need to be discussed in more depth. In one sense, low variance is not necessarily an issue, as it depends on the nature of the task - for node classification, if there is high homophily in labels, low variance is not necessarily bad. In other words, how low is “low variance” should perhaps be explained more clearly.

2) It looks to me the pre-diffusion step is critical in determining which channels have low variance. Is it always best to allow the diffusion to (nearly) converge or we control this in a more adaptive fashion?

3) Why choosing only one node to inject synthetic feature? Can it be selected in a more informative way than randomly? Also, what’s the impact of r (number of channels to inject synthetic features)?

4) Can the proposed method be tested on datasets with low label homophily? In my view this is when the proposed method might show clearest advantages over baselines such as FP.

---

### Official Review · Reviewer_FV8o · 2024-11-06

**Soundness:** 3
**Presentation:** 3
**Contribution:** 3
**Rating:** 6
**Confidence:** 5

**Summary:**

This paper investigates the limitations of diffusion-based graph imputation methods, particularly focusing on the issue of “low variance” channels. These channels contain imputed features with nearly identical values across nodes, leading to limited information for graph learning tasks. To address this problem, the authors propose a novel imputation method called FISF (Feature Imputation with Synthetic Features). FISF injects synthetic features into low-variance channels, increasing their variance and enhancing the distinctiveness of node representations. The paper presents empirical evidence and theoretical proofs demonstrating the effectiveness of FISF in various graph learning tasks, including semi-supervised node classification and link prediction, with varying missing rates.

**Strengths:**

1. The paper is well-written and well-structured, with clear explanations and figures that facilitate understanding.
2. The paper presents a novel approach to address the low-variance problem in diffusion-based graph imputation, which has not been explored extensively in previous work.
3. The paper provides strong empirical evidence and theoretical proofs to support its claims, making the contribution robust and reliable.
4. The proposed method, FISF, demonstrates superior performance in various graph learning tasks, making it a valuable tool for researchers and practitioners working with graphs containing missing features.

**Weaknesses:**

1. The complexity of the proposed method is confusing. For example, why the complexity contains O(|$\mathcal{E}$|) needs more clarification.
2. While the paper compares FISF with several existing methods, a more in-depth analysis and comparison with alternative methods, particularly in terms of scalability and computational efficiency, would strengthen the contribution. For example, authors can give running time comparisons on large graphs such as OGBN-Arxiv.
3. The performance discussion on heterophilic graphs is missing, while the competitor FP gives the analysis that diffusion based methods are not suitable for heterophilic graphs. The authors should clarify whether such a limitation still exists in FISF.

**Questions:**

1. In analysis, why does the authors say that the complexity of Dijkstra algorithm is O(n^2)? In fact, its complexity is O(nlogn) with a heap. Please clarify if a specific implementation of Dijkstra's algorithm is used that results in O(n^2) complexity, or if this is an error that needs correction?
2. How does the performance of FISF compare to other imputation methods in terms of scalability and computational efficiency?
3. Does the experimental dataset take the largest connected block? How does the method perform on non-fully connected datasets?
4. How does the performance of FISF on the heterophilic graphs?

---

### Note · Authors · 2024-11-15

I have read and agree with the venue's withdrawal policy on behalf of myself and my co-authors.